# Ca²⁺ channel and active zone protein abundance intersects with input-specific synapse organization to shape functional synaptic diversity

**Audrey T Medeiros[1], Scott J Gratz[2]\*, Ambar Delgado[2], Jason T Ritt[2,3], Kate M O'Connor-Giles[1,2,3]\***

[1]Neuroscience Graduate Training Program, Brown University, Providence, United States; [2]Department of Neuroscience, Brown University, Providence, United States; [3]Carney Institute for Brain Science, Brown University, Providence, United States

**\*For correspondence:**
scott_gratz@brown.edu (SJG);
kate_oconnor-giles@brown.edu
(KMO'C-G)

**Competing interest:** The authors declare that no competing interests exist.

**Abstract** Synaptic heterogeneity is a hallmark of nervous systems that enables complex and adaptable communication in neural circuits. To understand circuit function, it is thus critical to determine the factors that contribute to the functional diversity of synapses. We investigated the contributions of voltage-gated calcium channel (VGCC) abundance, spatial organization, and subunit composition to synapse diversity among and between synapses formed by two closely related *Drosophila* glutamatergic motor neurons with distinct neurotransmitter release probabilities ($P_r$). Surprisingly, VGCC levels are highly predictive of heterogeneous $P_r$ among individual synapses of either low- or high-$P_r$ inputs, but not between inputs. We find that the same number of VGCCs are more densely organized at high-$P_r$ synapses, consistent with tighter VGCC-synaptic vesicle coupling. We generated endogenously tagged lines to investigate VGCC subunits in vivo and found that the α2δ–3 subunit Straightjacket along with the CAST/ELKS active zone (AZ) protein Bruchpilot, both key regulators of VGCCs, are less abundant at high-$P_r$ inputs, yet positively correlate with $P_r$ among synapses formed by either input. Consistently, both Straightjacket and Bruchpilot levels are dynamically increased across AZs of both inputs when neurotransmitter release is potentiated to maintain stable communication following glutamate receptor inhibition. Together, these findings suggest a model in which VGCC and AZ protein abundance intersects with input-specific spatial and molecular organization to shape the functional diversity of synapses.

## eLife assessment

Calcium channels are key regulators of synaptic strength and plasticity. The authors generate new endogenous tags of the *Drosophila* channel Cac as well as auxiliary subunits to investigate distinct calcium channel functions at the fly NMJ, Is and Ib. They demonstrate functions for voltage-gated calcium channel subunits in promoting synaptic strength, diversity, and plasticity with a series of **convincing** analyses. The work is **important** and has broad implications. In addition, the newly developed tools should be quite beneficial for fly biologists.

## Introduction

The broad and complex functions of neural circuits depend on diverse neuronal subtypes communicating through synapses with distinct properties. Thus, understanding how synaptic diversity is established is critical for understanding neural circuit function. Neurotransmission occurs at specialized

membranes called active zones (AZs) where action potentials drive the opening of voltage-gated $Ca^{2+}$ channels (VGCCs) to trigger $Ca^{2+}$-dependent synaptic vesicle (SV) fusion and neurotransmitter release. Neurotransmitter release properties are determined locally at individual synapses and vary considerably between neuronal subtypes and within homogeneous populations of neurons (*Ariel et al., 2012*; *Atwood and Karunanithi, 2002*; *Branco and Staras, 2009*; *Hatt and Smith, 1976*). In fact, functional imaging studies in *Drosophila* demonstrate that even single neurons forming synapses with the same postsynaptic partner display heterogeneous synaptic strength among individual AZs (*Guerrero et al., 2005*; *Melom et al., 2013*; *Peled and Isacoff, 2011*).

Presynaptic strength is defined as the likelihood of neurotransmitter release following an action potential (probability of release, $P_r$). This probabilistic process is determined by the number of functional SV release sites and their individual probability of vesicle release. The probability of SV release is highly dependent on transient increases in intracellular $Ca^{2+}$ levels at vesicular sensors. Accordingly, SV release sites and VGCCs are key substrates for generating diversity of synaptic function (*Akbergenova et al., 2018*; *Aldahabi et al., 2022*; *Chen et al., 2015*; *Fedchyshyn and Wang, 2005*; *Fekete et al., 2019*; *Gratz et al., 2019*; *Holderith et al., 2012*; *Laghaei et al., 2018*; *Miki et al., 2017*; *Nakamura et al., 2015*; *Newman et al., 2022*; *Rebola et al., 2019*; *Reddy-Alla et al., 2017*; *Sauvola et al., 2021*; *Sheng et al., 2012*). Numerous studies have demonstrated that VGCC abundance is highly correlated with $P_r$ across species (*Akbergenova et al., 2018*; *Gratz et al., 2019*; *Holderith et al., 2012*; *Miki et al., 2017*; *Nakamura et al., 2015*; *Sheng et al., 2012*). Paradoxically, this is not always the case. For example, a recent study investigated two cerebellar synaptic subtypes, one high-$P_r$ formed by inhibitory stellate cells and one low-$P_r$ formed by excitatory granule cells, and found higher VGCC levels at low-$P_r$ granule synapses (*Rebola et al., 2019*). Since VGCCs in closer proximity to release sites are expected to have a greater impact on vesicular release probability than those positioned farther away, the spatial coupling of VGCCs and SVs at AZs is a critical determinant of $P_r$ (*Chen et al., 2015*; *Eggermann et al., 2011*; *Fedchyshyn and Wang, 2005*; *Nakamura et al., 2015*; *Rebola et al., 2019*). Indeed, at high-$P_r$ stellate synapses, a 'perimeter release' AZ organization places VGCCs ~40 nm closer to SVs than at low-$P_r$ granular synapses (*Rebola et al., 2019*). Another recent study investigated two functionally distinct connections formed by CA1 pyramidal cells (*Aldahabi et al., 2022*). While $Ca^{2+}$ influx was higher at the high-$P_r$ synapse, raising $Ca^{2+}$ influx at the low-$P_r$ synapse to match the high-$P_r$ synapse did not equalize $P_r$.

To further investigate this paradox, we sought a system where we could investigate the relationship between VGCCs and $P_r$ both within and between two closely related neurons that form synapses with distinct release probabilities. *Drosophila* muscles are innervated by two glutamatergic motor neurons, one tonic and one phasic, that form type Ib and type Is synapses, respectively. Type Ib synapses have relatively low $P_r$ and facilitate, whereas type Is synapses have higher $P_r$ and depress in response to high-frequency stimulation (*Aponte-Santiago et al., 2020*; *Lnenicka and Keshishian, 2000*). In this study, we investigated how VGCC abundance, spatial organization, and subunit composition contribute to synaptic heterogeneity at AZs of low-$P_r$ type Ib and high-$P_r$ type Is inputs to the same postsynaptic targets. We find that individual synapses formed by both low- and high-$P_r$ inputs exhibit heterogeneous release properties that can be predicted by VGCC abundance alone. However, VGCC abundance does not correspond to differences in $P_r$ between the two inputs. We identify underlying molecular and organizational differences that may alter the relationship between VGCC abundance and $P_r$ at low- vs. high-$P_r$ inputs. We further find that the homeostatic potentiation of neurotransmitter release triggered by glutamate receptor inhibition involves dynamic increases in VGCCs, the α2δ–3 subunit Straightjacket (Stj), and the AZ cytomatrix protein Bruchpilot (Brp) across AZs of both inputs. These findings provide insight into how VGCC and AZ protein abundance intersects with underlying molecular and organizational differences between inputs to contribute to greater synaptic diversity.

## Results
### VGCC levels predict $P_r$ within, but not between, inputs

To investigate the relationship between VGCC levels and neurotransmitter release properties at functionally distinct synapses, we took advantage of the two motor neuron subtypes with low and high release probabilities that innervate most *Drosophila* muscles (*Aponte-Santiago and Littleton, 2020*; *Kurdyak et al., 1994*). These glutamatergic neuromuscular junctions (NMJs) contain hundreds of

individual synapses that are accessible to single AZ functional imaging using genetically encoded Ca$^{2+}$ indicators.

In *Drosophila*, Cacophony (Cac) is the sole Ca$_v$2 pore-forming subunit and is the VGCC responsible for triggering synaptic transmission (*Kawasaki et al., 2000*; *Macleod et al., 2006*; *Peng and Wu, 2007*; *Smith et al., 1996*). To simultaneously monitor neurotransmitter release and VGCC levels, we swapped the N-terminal sfGFP tag in our well-characterized *cac$^{sfGFP-N}$* line for a Td-Tomato tag (*cac$^{Td-Tomato-N}$*) and confirmed that the tag does not impair synaptic function (*Figure 1—figure supplement 1*; *Figure 1A–F*; *Ghelani et al., 2023*; *Gratz et al., 2019*). We then expressed postsynaptically targeted GCaMP6f (SynapGCaMP6f; *Newman et al., 2017*), which reports Ca$^{2+}$ influx through glutamate receptors in response to neurotransmitter release, in *cac$^{Td-Tomato-N}$* animals for a plus/minus readout. We and others have previously shown that Cac levels are highly predictive of P$_r$ at individual type Ib AZs (*Akbergenova et al., 2018*; *Gratz et al., 2019*). To determine if VGCC levels are similarly predictive at high-P$_r$ type Is AZs, we measured Cac$^{Td-Tomato-N}$ fluorescence intensity and monitored neurotransmitter release in response to 0.2 Hz stimulus at individual synapses. To enable direct comparisons between the two inputs, we simultaneously imaged type Ib and Is synapses at NMJ 6/7 (*Figure 1A*). We quantified the number of times a vesicle was released over 120 stimuli to determine single-synapse P$_r$. As has been previously reported, we found that type Is synapses exhibited significant heterogeneity and higher average P$_r$ than type Ib synapses (*Figure 1B and C*; *Lu et al., 2016*; *Newman et al., 2022*). Consistent with their higher P$_r$, type Is connections contain relatively fewer low-P$_r$ and more high-P$_r$ AZs (*Figure 1D*). We next investigated the correlation between P$_r$ and VGCC levels and found that at type Is inputs, single-AZ Cac intensity positively correlates with P$_r$ (*Figure 1E*). We also observe a strong positive correlation between VGCC levels and P$_r$ at type Ib inputs to the same muscles (*Figure 1F*), consistent with our and others' prior findings (*Akbergenova et al., 2018*; *Gratz et al., 2019*; *Newman et al., 2022*).

A simple prediction of the observation that VGCC levels correlate highly with P$_r$ at individual AZs of both low- and high-P$_r$ inputs is that Cac levels will be higher at synapses of type Is inputs than type Ib. We analyzed Cac$^{sfGFP-N}$ levels at individual type Ib and Is synapses and found that average Cac levels are the same at type Ib and Is AZs (*Figure 1G and H*). Cac levels are also similarly distributed across AZs of the two inputs (*Figure 1I*). Together, these findings indicate that the relationship between VGCC levels and P$_r$ differs between the two inputs. Consistently, when we directly compare the best-fit lines for the relationship between Cac levels and P$_r$ at type Ib and Is inputs from our correlative functional imaging data (*Figure 1E and F*), we find that the slopes are significantly different (*Figure 1J*). Across type Is AZs, a similar range of VGCC levels supports a higher range of release probabilities. Thus, VGCCs can predict P$_r$ within synaptic subtypes, but not between AZs of different synaptic subtypes, providing a framework for understanding seemingly contradictory findings on the role of VGCCs in determining P$_r$.

## VGCC clusters are more compact at AZs of high-P$_r$ type Is inputs

Many differences between low-P$_r$ type Ib and high-P$_r$ type Is AZs have been described (*Aponte-Santiago and Littleton, 2020*; *Aponte-Santiago et al., 2020*; *Atwood et al., 1993*; *He et al., 2023*; *Jetti et al., 2023*; *Kurdyak et al., 1994*; *Lu et al., 2016*; *Medeiros and O'Connor-Giles, 2023*). Perhaps most notably, type Is AZs experience ~twofold greater Ca$^{2+}$ influx than type Ib (*He et al., 2023*; *Lu et al., 2016*). While this alone could explain the estimated 3-fold greater P$_r$ at type Is AZs and is certainly a key factor, several lines of evidence argue for additional contributors. A recent study using a botulinum transgene to isolate type Ib and Is synapses for electrophysiological analysis found that increasing external [Ca$^{2+}$] from physiological levels (1.8 mM) to 3 mM or even 6 mM does not result in a 3-fold increase in EPSCs or quantal content at type Ib synapses and type Ib synapses continue to facilitate at 3 mM external [Ca$^{2+}$] (*He et al., 2023*). Using this approach, they further found that type Ib synapses are more sensitive to the slow Ca$^{2+}$ chelator EGTA, indicating looser VGCC-SV coupling.

We investigated the spatial distribution of VGCCs at type Ib and Is AZs using 3D dSTORM single-molecule localization microscopy (SMLM). An individual VGCC complex is estimated to be ~10 nm in diameter with the most common immunolabeling techniques adding significantly to their size and creating a linkage error of ~20 nm between the target molecule and fluorescent reporter (*Früh et al., 2021*; *Liu et al., 2022*; *Thomas, 2000*). For following VGCC dynamics using single-particle

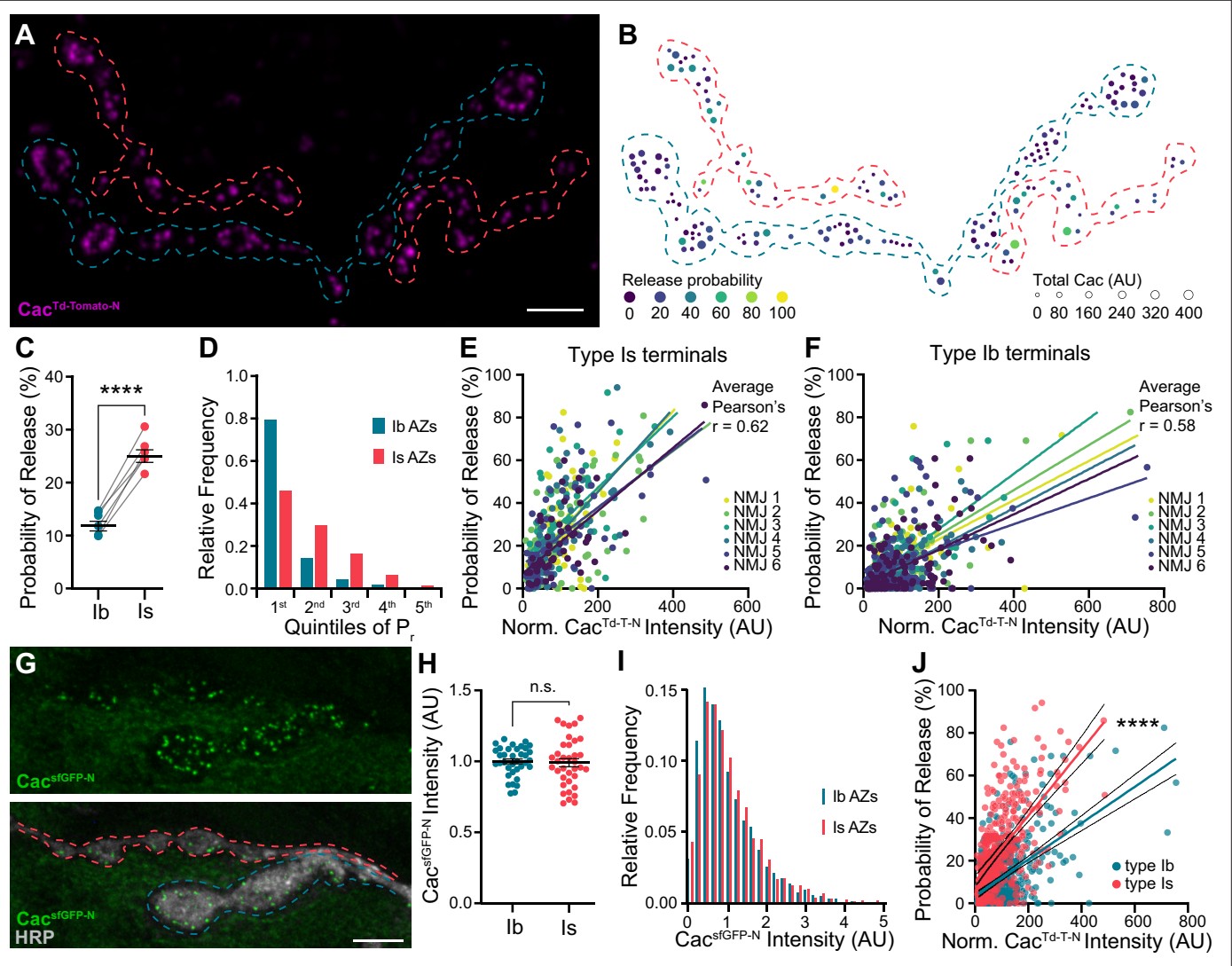

**Figure 1.** VGCC levels predict $P_r$ within, but not between, inputs. (**A**) Representative confocal Z-projection of Cac[Td-Tomato-N] (magenta) with type Ib (blue) and type Is (red) terminals outlined. (**B**) AZ heat map of terminals in A with color indicating $P_r$ and size representing sum Cac intensity levels in arbitrary units (AU). (**C**) Average single-AZ probability of release at type Ib and Is terminals. N=6 animals, 6 NMJs. (**D**) Quintile distribution of single-AZ $P_r$ frequency at type Ib and Is inputs. (**E, F**) Correlation between normalized Cac[Td-Tomato-N] intensity and $P_r$ at type Is and Ib AZs of the same 6 NMJs. Each dot represents a single AZ and each color corresponds to an individual NMJ with linear regression lines indicated for each. (**G**) Top, representative confocal Z-projection of Cac[sfGFP-N]. Bottom, Cac[sfGFP-N] in green with HRP marking neuronal membranes in gray. Type Ib (blue) and type Is (red) terminals are outlined. (**H**) Quantification of Cac[sfGFP-N] AZ intensity at type Ib and Is terminals. Each data point represents the average normalized single AZ sum intensity for an individual NMJ. (**I**) Distribution of normalized Cac[sfGFP-N] intensity from single type Ib and Is AZs in H (X-axis cutoff at 5.0). (**J**) Comparison between normalized Cac[Td-Tomato-N] and $P_r$ of type Ib and Is AZs combined from E-F with linear regression lines (blue and red, respectively) and 95% confidence intervals (black lines) indicated. All scale bars = 5 µm, all error bars indicate S.E.M, ****p<0.0001; ns, not significant. N's, absolute values, and statistical information is detailed in **Supplementary file 1a**.

The online version of this article includes the following figure supplement(s) for figure 1:

**Figure supplement 1.** Electrophysiological validation of endogenously tagged cacophony lines.

(**A-C**) Representative traces of EJPs (top) and mEJPs (bottom) in control, cacHaloTag-N, and cacTd-Tomato-N. (D-F) Quantification of EJPs, mEJPs, and quantal content (QC). All error bars indicate S.E.M., ns, not significant.

tracking via photoactivation localization microscopy (sptPALM), we recently incorporated mEOS4b (**Paez-Segala et al., 2015**) at the same N-terminal site we previously used to endogenously tag Cac, achieving a linkage error of less of than 5 nm (**Ghelani et al., 2023**; **Gratz et al., 2019**). To gain more flexibility in labeling Cac without adding to the linkage error, we swapped the mEOS tag for a similarly sized HaloTag (*cac[HaloTag-N]*). *cac[HaloTag-N]* flies are fully viable, do not display significant defects in

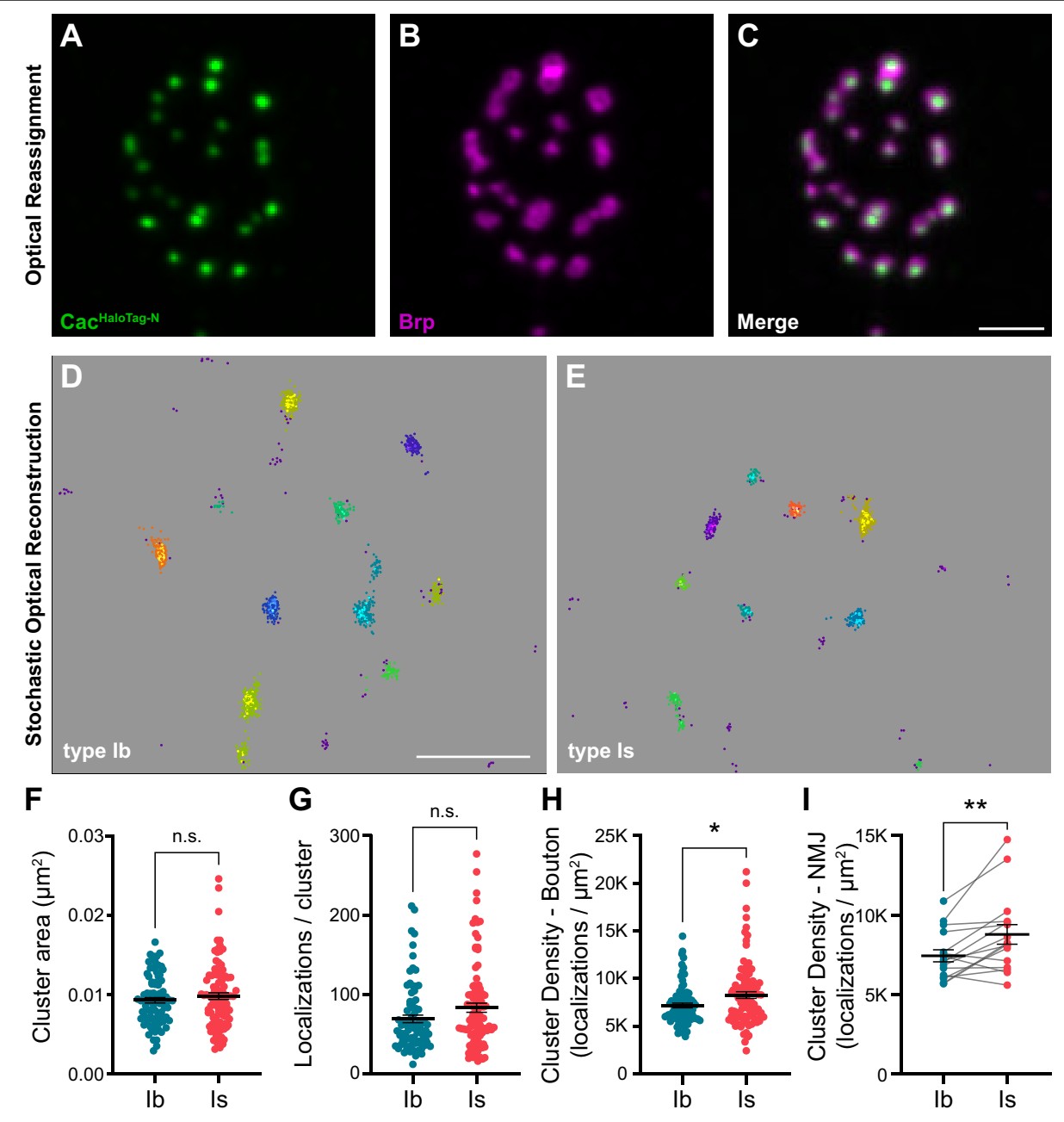

**Figure 2.** VGCC clusters are more compact at AZs of high-P_r type Is inputs. (**A–C**) Representative SoRa Z-projection of Cac^HaloTag-N (green), Brp (magenta), and merge. (**D, E**) Representative boutons of STORM Cac^HaloTag-N clusters as identified by DBSCAN at type Ib and Is boutons as indicated. Each color represents an individual identified cluster with purple scattered dots identifying excluded background signal. (**F–H**) Analysis of STORM-acquired Cac^HaloTag-N clusters where each data point represents the respective single-cluster measurement averaged over individual boutons. (**F**) Quantification of Cac^HaloTag-N cluster area at type Ib and Is AZs. (**G**) Quantification of localizations per cluster at type Ib and Is boutons. (**H**) Calculated Cac^HaloTag-N cluster density at type Ib and Is AZs. (**I**) Paired analysis of calculated AZ cluster density averaged over individual type Ib and Is inputs to the same muscle. All scale bars = 1 μm, all error bars indicate S.E.M. **p<0.01; *p<0.05; ns, not significant. N's, absolute values, and statistical information is detailed in **Supplementary file 1a**.

synaptic function, and exhibit normal Cac localization at AZs as observed in super-resolution optical reassignment images, where Brp is arranged in rings surrounding puncta of VGCCs (**Figure 1—figure supplement 1**; **Figure 2A–C**, **Ghelani et al., 2023**). HaloTag, which covalently binds synthetic ligands, is 3.3 nm in diameter (**Los et al., 2008**; **Yazaki et al., 2020**), yielding a linkage error well under 5 nm. cac^HaloTag-N larvae were stained with JaneliaFluor646 HaloTag ligand (**Grimm et al., 2015**) and

horseradish peroxidase (HRP) to distinguish between type Ib and Is branches and enable simultaneous imaging of the two inputs at a single NMJ. We then used density-based spatial clustering of applications with noise (DBSCAN) analysis to identify Cac clusters at type Ib and Is AZs (*Figure 2D and E*; *Ehmann et al., 2014*). We find that the average size of Cac$^{HaloTag-N}$ clusters is similar at low- and high-P$_r$ AZs (*Figure 2F*), with mean diameters of approximately 102 nm and 105 nm, respectively. This is similar to the Cac$^{mEOS4b-N}$ type Ib cluster size observed by sptPALM imaging (*Ghelani et al., 2023*). In agreement with our confocal level data, the number of localizations per cluster was similar at low- and high-P$_r$ AZs (*Figure 2G*). We then calculated the average Cac density per AZ and found that VGCCs are significantly more densely organized at high-P$_r$ type Is AZs than low-P$_r$ type Ib AZs (*Figure 2H, I*). Greater VGCC AZ density at type Is AZs is consistent with a recent SMLM study using antibodies to label Cac, indicating these results are robust to different labeling approaches (*Newman et al., 2022*). Together, these findings suggest that more compact organization of VGCCs increases their coupling to SVs and contributes to the steeper relationship between VGCC levels and P$_r$ at high-P$_r$ type Is AZs.

## Differences in Bruchpilot levels and function at low- and high-P$_r$ inputs

To understand how these nanoscale differences in VGCC organization might be established, we investigated the AZ scaffolding protein Brp. Brp/CAST/ELKS family proteins function as central organizers of both VGCCs and SV release sites at developing synapses (*Dai et al., 2006*; *Dong et al., 2018*; *Hallermann et al., 2010*; *Held et al., 2016*; *Kittel et al., 2006*; *Liu et al., 2014*; *McDonald et al., 2020*; *Radulovic et al., 2020*). Like Cac, Brp is more densely arranged at type Is AZs as measured through SMLM and stimulated emission depletion (STED) imaging studies (*He et al., 2023*; *Jetti et al., 2023*; *Mrestani et al., 2021*). We simultaneously imaged type Ib and Is inputs and found lower Brp levels at type Is AZs (*Figure 3A and B*). Since Cac levels are similar at AZs of the two inputs, lower Brp levels result in a significantly higher Cac:Brp ratio at type Is synapses, which we hypothesize promotes compact organization of VGCCs (*Figure 3C*). In contrast, we and others have previously shown that Brp levels positively correlate with P$_r$ among AZs of low-P$_r$ type Ib inputs (*Gratz et al., 2019*; *Muhammad et al., 2015*; *Newman et al., 2017*; *Peled et al., 2014*; *Reddy-Alla et al., 2017*). Consistently, Brp and Cac levels strongly correlate at type Ib AZs (*Gratz et al., 2019*) and we observe a similarly strong correlation across individual type Is AZs (*Figure 3D*). Thus, like VGCCs, Brp levels contribute in distinct ways to synaptic heterogeneity within vs. between low- and high-P$_r$ inputs, likely due to differences in AZ organization between the two synaptic subtypes.

We next investigated the requirement for Brp in promoting VGCC accumulation at low- and high-P$_r$ inputs by analyzing Cac$^{sfGFP-N}$ levels in *brp* null mutants (*brp$^{-/-}$*; *Figure 3E and F*). Cac$^{sfGFP-N}$ levels are diminished at both type Ib and Is AZs, demonstrating a conserved role for Brp in promoting Cac accumulation at both inputs (*Figure 3G*). The relative decrease in Cac levels at type Ib AZs is significantly greater than at type Is AZs, indicating a greater requirement for Brp in regulating VGCC levels at low-P$_r$ type Ib synapses (*Figure 3H*). This suggests that an additional factor or factors function with or upstream of Brp to establish differences in Brp dependence at low- and high-P$_r$ AZs.

## Brp differentially regulates VGCC dynamics at low- and high-P$_r$ synapses during presynaptic homeostatic potentiation

In response to acute or chronic inhibition of glutamate receptors at NMJs, *Drosophila* motor neurons homeostatically increase neurotransmitter release to maintain synaptic communication (*Davis and Müller, 2015*; *Frank, 2014*; *James et al., 2019*). Pharmacological inhibition of glutamate receptors with the wasp toxin Philanthotoxin-433 (PhTx) induces acute presynaptic homeostatic potentiation of release (PHP) within minutes (*Frank et al., 2006*). We and others have demonstrated that acute PHP involves rapid changes in VGCC and other AZ protein levels at type Ib AZs (*Böhme et al., 2019*; *Gratz et al., 2019*; *Weyhersmüller et al., 2011*). Recent studies have revealed significant differences in the induction of PHP at low- and high-P$_r$ synaptic inputs under different conditions (*Genç and Davis, 2019*; *Newman et al., 2017*; *Sauvola et al., 2021*). PhTx induces acute PHP at both type Ib and Is synapses (*Genç and Davis, 2019*), but the molecular changes underlying PHP at high-P$_r$ type Is AZs remain unknown. To compare the dynamic modulation of VGCCs at low- and high-P$_r$ AZs, we treated *cac$^{sfGFP-N}$* larvae with non-saturating concentrations of PhTx for 10 min, then quantified Cac and Brp levels at type Ib and Is AZs (*Figure 4A and B*). We observe a significant PhTx-induced increase in Brp and Cac$^{sfGFP-N}$ levels at type Is AZs similar to type Ib (*Figure 4C and D*; *Gratz et al., 2019*). Thus,

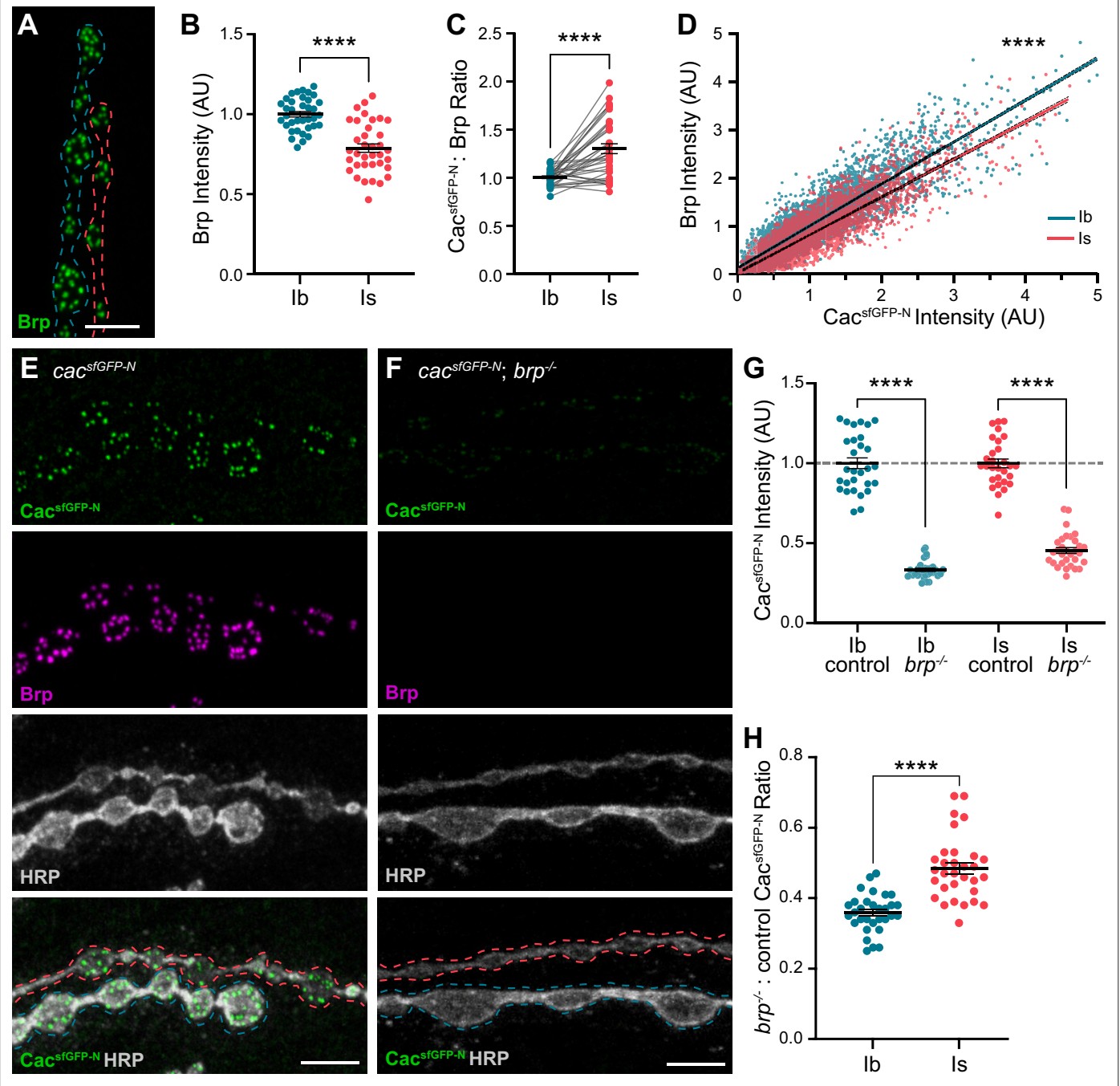

**Figure 3.** Differences in Bruchpilot (Brp) levels and function at low-and high-P$_r$ inputs. (**A**) Representative confocal Z-projection of Brp expression at type Ib (blue outline) and type Is (red outline) terminals. (**B**) Quantification of Brp AZ intensity at type Ib and Is terminals. (**C**) Ratio of normalized Cac$^{sfGFP-N}$:Brp levels at type Ib and Is inputs to the same muscles. (**D**) Correlation of Cac$^{sfGFP-N}$ and Brp at type Ib and Is single AZs with linear regression lines (blue and red, respectively) and 95% confidence intervals (black dotted lines) indicated. (**E, F**) Representative confocal Z-projections of Cac$^{sfGFP-N}$ (green), Brp (magenta), HRP (white), and merge at type Ib (blue outline) and Is (red outline) terminals of *cac$^{sfGFP-N}$* (control) or *cac$^{sfGFP-N}$;brp$^{-/-}$* (*brp$^{-/-}$*) animals. (**G**) Quantification of Cac$^{sfGFP-N}$ normalized fluorescence intensity at type Ib and Is AZs of control vs *brp$^{-/-}$* NMJs. (**H**) Ratio of Cac$^{sfGFP-N}$ fluorescence intensity at type Ib and Is AZs between *brp$^{-/-}$*and control NMJs. For B and G, each data point represents the average normalized single AZ sum intensity for an individual NMJ. All scale bars = 5 µm, all error bars indicate S.E.M., ****p<0.0001. N's, absolute values, and statistical information is detailed in *Supplementary file 1a*.

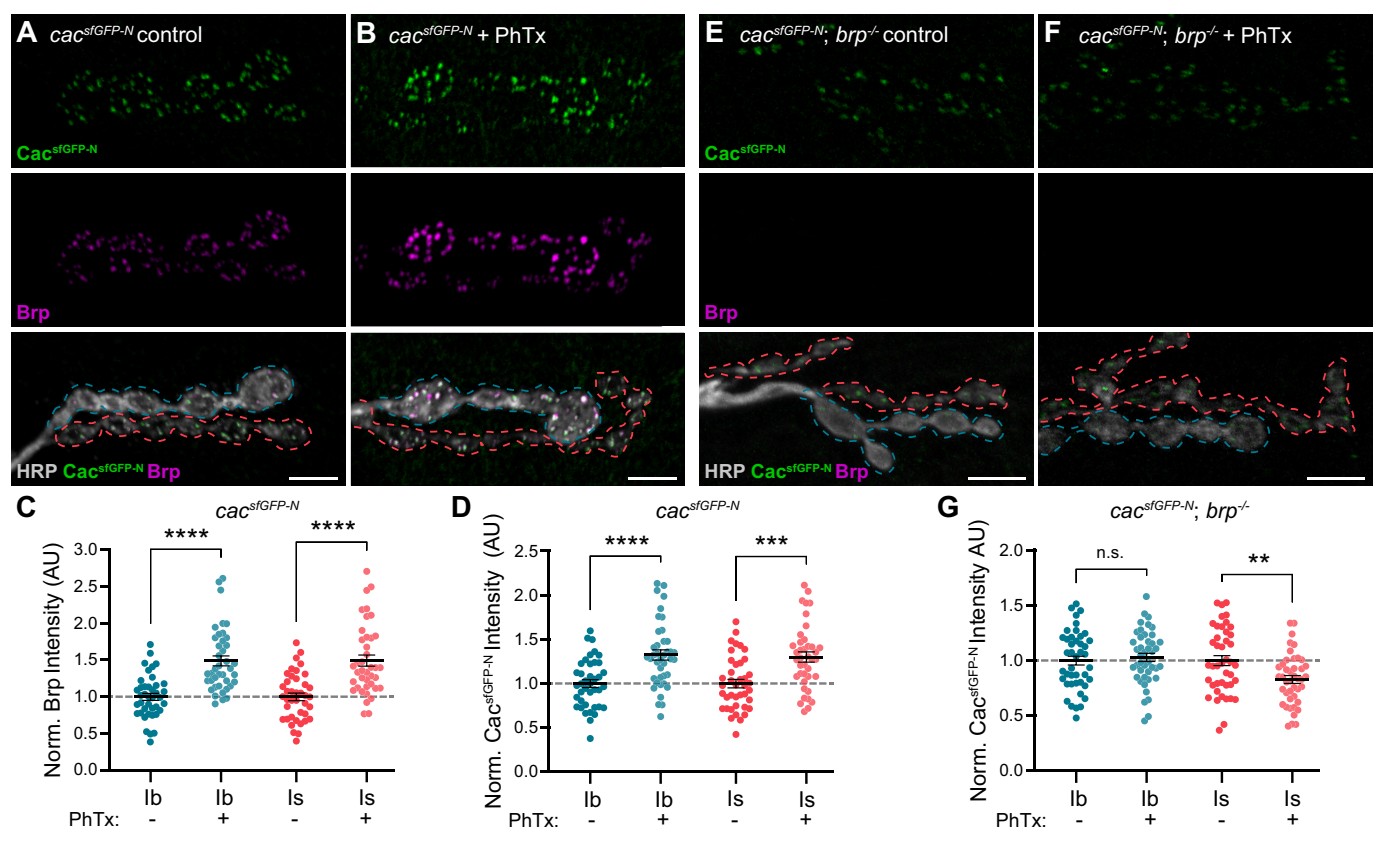

**Figure 4.** Brp differentially regulates VGCC dynamics at low-and high-$P_r$ inputs during *presynaptic homeostatic potentiation*. (**A, B**) Representative confocal Z-projections of Cac[sfGFP-N] (top, green), Brp (middle, magenta), and both merged with HRP (bottom, gray) at untreated and PhTx-treated *cac[sfGFP-N]* NMJs showing type Ib (blue) and type Is (red) terminals. (**C**) Quantification of Brp fluorescence intensity at untreated and PhTx-treated type Ib and Is terminals. (**D**) Quantification of Cac[sfGFP-N] fluorescence intensity at untreated and PhTx-treated type Ib and Is terminals. (**E, F**) Representative confocal Z-projections of Cac[sfGFP-N] (top, green), Brp (middle, magenta), and both merged with HRP (bottom, gray) at untreated and PhTx-treated *cac[sfGFP-N];brp[-/-]* NMJs showing type Ib (blue) and type Is (red) terminals. (**G**) Quantification of Cac[sfGFP-N] fluorescence intensity at untreated and PhTx-treated *cac[sfGFP-N];brp[-/-]* type Ib and Is terminals. For all quantifications, each data point represents the average normalized single AZ sum intensity for an individual NMJ. All scale bars = 5 µm, all error bars indicate S.E.M. ****p<0.0001; ***p<0.001; **p<0.01; ns, not significant. N's, absolute values, and statistical information is detailed in *Supplementary file 1a*.

despite their distinct baseline transmission and organizational properties, PhTx-induced potentiation of neurotransmitter release involves the rapid accumulation of VGCCs at both low- and high-$P_r$ AZs.

At low-$P_r$ type Ib AZs, Brp is a critical regulator of PHP-induced accumulation of proteins associated with SV priming and release, specifically Unc13A and Syntaxin-1A (*Böhme et al., 2019*). At type Ib AZs, PhTx also induces a Brp-dependent increase in Cac density and decrease in channel mobility (*Ghelani et al., 2023*). Notably, Brp itself becomes more densely organized during PHP (*Ghelani et al., 2023*), consistent with its denser organization at high-$P_r$ type Is AZs (*Mrestani et al., 2021*). Since baseline accumulation of VGCCs depends less on Brp at high-$P_r$ type Is AZs, we investigated the role of Brp in promoting dynamic increases in VGCC levels at type Ib and Is AZs by treating *cac[sfGFP-N]; brp[-/-]* larvae with PhTx followed by quantification of Cac[sfGFP-N] levels (*Figure 4E and F*). We find that PhTx failed to induce accumulation of Cac at either type Ib or Is AZs in *brp[-/-]* mutants, demonstrating a shared requirement for Brp in regulating VGCC dynamics at both inputs (*Figure 4G*). In contrast to no change at type Ib AZs, Cac[sfGFP-N] levels are significantly decreased at type Is AZs (*Figure 4G*), revealing an input-specific role for Brp in maintaining VGCC levels during the dynamic reorganization of AZs. Consistently, *Ghelani et al., 2023* found that whereas PhTx induces a decrease in Cac mobility at wild-type type Ib AZs, in *brp[-/-]* mutants Cac mobility increases (*Ghelani et al., 2023*). Together, these findings suggest potentiating synapses must coordinate the accumulation of new VGCCs with

the stabilization of existing channels, and that meeting this challenge is more dependent upon Brp at high-$P_r$ AZs.

## Endogenous tagging of VGCC auxiliary subunits reveals distinct synaptic expression patterns

In addition to the pore-forming α subunits, VGCCs comprise auxiliary α2δ and β subunits that regulate forward channel trafficking, membrane insertion, and function (*Figure 5A*; *Campiglio and Flucher, 2015*; *Dolphin and Lee, 2020*; *Weiss and Zamponi, 2017*). β subunits interact with pore-forming α subunits intracellularly, whereas GPI-anchored α2δ subunits are largely extracellular. Beyond their interaction with α subunits, α2δs have been shown to interact with a growing number of extracellular proteins to promote synaptogenesis (*Bauer et al., 2010*; *Dolphin, 2018*; *Risher et al., 2018*). The *Drosophila* genome encodes one synaptic Ca$_v$2 α subunit (Cac), one β subunit, and three α2δ subunits (*Littleton and Ganetzky, 2000*). Auxiliary subunits are both spatially and temporally regulated and broadly able to interact with α subunits. Thus, the subunit composition of channel complexes is a potential source of significant diversity in both the spatial and functional regulation of VGCCs.

*Ca-β* encodes the sole *Drosophila* β subunit and has been shown to enhance Ca$^{2+}$ transients in sensory neurons (*Kanamori et al., 2013*). *Drosophila* α2δ–3, also known as Straightjacket (Stj), has well-characterized roles at the NMJ in promoting Ca$^{2+}$ channel clustering, homeostatic plasticity, and, independently of Cac, synapse formation and organization (*Dickman et al., 2008*; *Hoover et al., 2019*; *Kurshan et al., 2009*; *Ly et al., 2008*; *Schöpf et al., 2021*; *Wang et al., 2016*). While low sequence homology between α2δ subunits within and across species makes 1:1 mapping difficult, the remaining two *Drosophila* α2δs map more closely to mammalian α2δ–3 and –4 than α2δ–1 and –2. Stolid was recently shown to promote dendritic Cac expression in motor neurons, whereas Ma2d is known to function in muscle where it is broadly expressed (*Heinrich and Ryglewski, 2020*; *Reuveny et al., 2018*). The synaptic localization of endogenous auxiliary subunits with VGCCs remains unknown in *Drosophila*.

To explore potential differences in VGCC subunit composition at type Ib and Is synapses, we used CRISPR gene editing to incorporate endogenous V5 tags in sequence common to all isoforms of *stj*, *stolid,* and *Ca-β* (*Bruckner et al., 2017*; *Gratz et al., 2014*). We inserted V5 after the N-terminal signal peptides of Stj and Stolid and near the C-terminus of Ca-β (see Materials and methods for details), and confirmed that the incorporation of the peptide tags did not impair neurotransmission (*Figure 5B–E*). We investigated the expression of each endogenously tagged subunit in the larval ventral ganglion and found that all subunits are expressed in the synaptic neuropil in a pattern similar to the α subunit Cac (*Figure 5F–H*; *Gratz et al., 2019*). Similar to Cac, Ca-β$^{V5-C}$ is highly enriched in the mushroom bodies of the larval brain. We next investigated expression at the larval NMJ where Cac localizes in a single punctum at each AZ and found that only Ca-β$^{V5-C}$ and Stj$^{V5-N}$ are present (*Figure 5F–H*). This aligns with the recent finding that Stolid does not play a role in regulating Ca$^{2+}$ transients at the larval NMJ (*Heinrich and Ryglewski, 2020*). We also observe Ca-β$^{V5-C}$ expression in muscle as expected for the sole *Drosophila* β subunit (*Figure 5F* and see *Figure 6C*).

## Stj/α2δ-3 levels are lower at AZs of high-$P_r$ type Is inputs

To investigate Ca-β$^{V5-C}$ and Stj$^{V5-N}$ localization at type Ib and Is AZs, we used super-resolution optical reassignment microscopy. Both subunits localize to AZs labeled with Cac or the CAST/ELKS AZ cytomatrix protein Brp (*Figure 6A and B*). We observe Brp rings surrounding puncta of VGCCs including Ca-β$^{V5-C}$ (*Figure 6A*). The tight localization of both subunits to central AZ puncta suggests they are associated with α subunits and predicts that Ca-β$^{V5-C}$ and Stj$^{V5-N}$ levels, like Cac, will be similar at the low- and high-$P_r$ synapses. To test this, we imaged Ca-β$^{V5-C}$ and Stj$^{V5-N}$ levels at both inputs simultaneously using confocal microscopy and measured fluorescence intensity (*Figure 6C and D*). As predicted, we found that Ca-β$^{V5-C}$ levels are similar at type Ib and Is AZs (*Figure 6E*). In contrast, Stj$^{V5-N}$ levels are significantly lower at high-$P_r$ type Is AZs (*Figure 6F*). Thus, while Cac and Ca-β are present in similar ratios at AZs of both inputs, surprisingly, the same is not true of Stj/α2δ–3 with high-$P_r$ type Is AZs exhibiting lower levels of Stj. This unexpected finding indicates that α:α2δ–3 stoichiometry is not always 1:1 in vivo and differs at low- and high-$P_r$ synapses. This is consistent with studies of mammalian subunits indicating that in contrast to β subunits, α2δ interactions with α subunits may be transient,

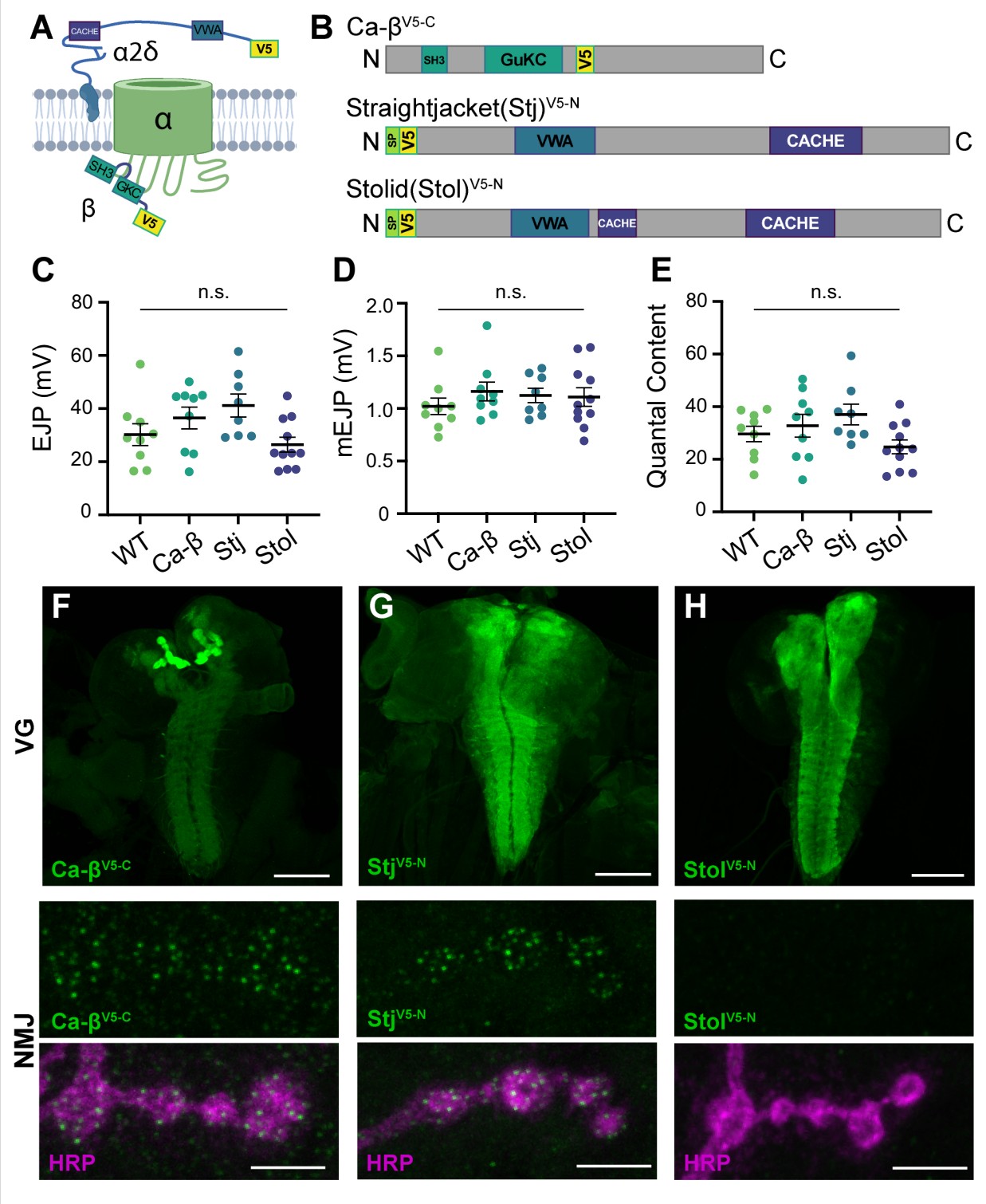

**Figure 5.** Endogenous tagging of VGCC auxiliary subunits reveals distinct synaptic expression patterns. (**A**) Schematic of a $Ca^{2+}$ channel complex with tagged auxiliary subunits (created with BioRender). (**B**) Schematic of Ca-β (isoform PL shown), Stj (isoform PC), and Stolid (isoform H/I) indicating endogenous tag locations. (**C–E**) Quantification of EJPs, mEJPs, and quantal content for each endogenously tagged line. (**F–H**) Representative confocal Z-projections of auxiliary subunit expression (green) at the larval ventral ganglion (VG, top, scale bars = 100 μm) and NMJs co-labeled with anti-HRP (magenta, bottom, scale bars = 5 μm). All error bars indicate S.E.M., ns, not significant. N's, absolute values, and statistical information is detailed in *Supplementary file 1a*.

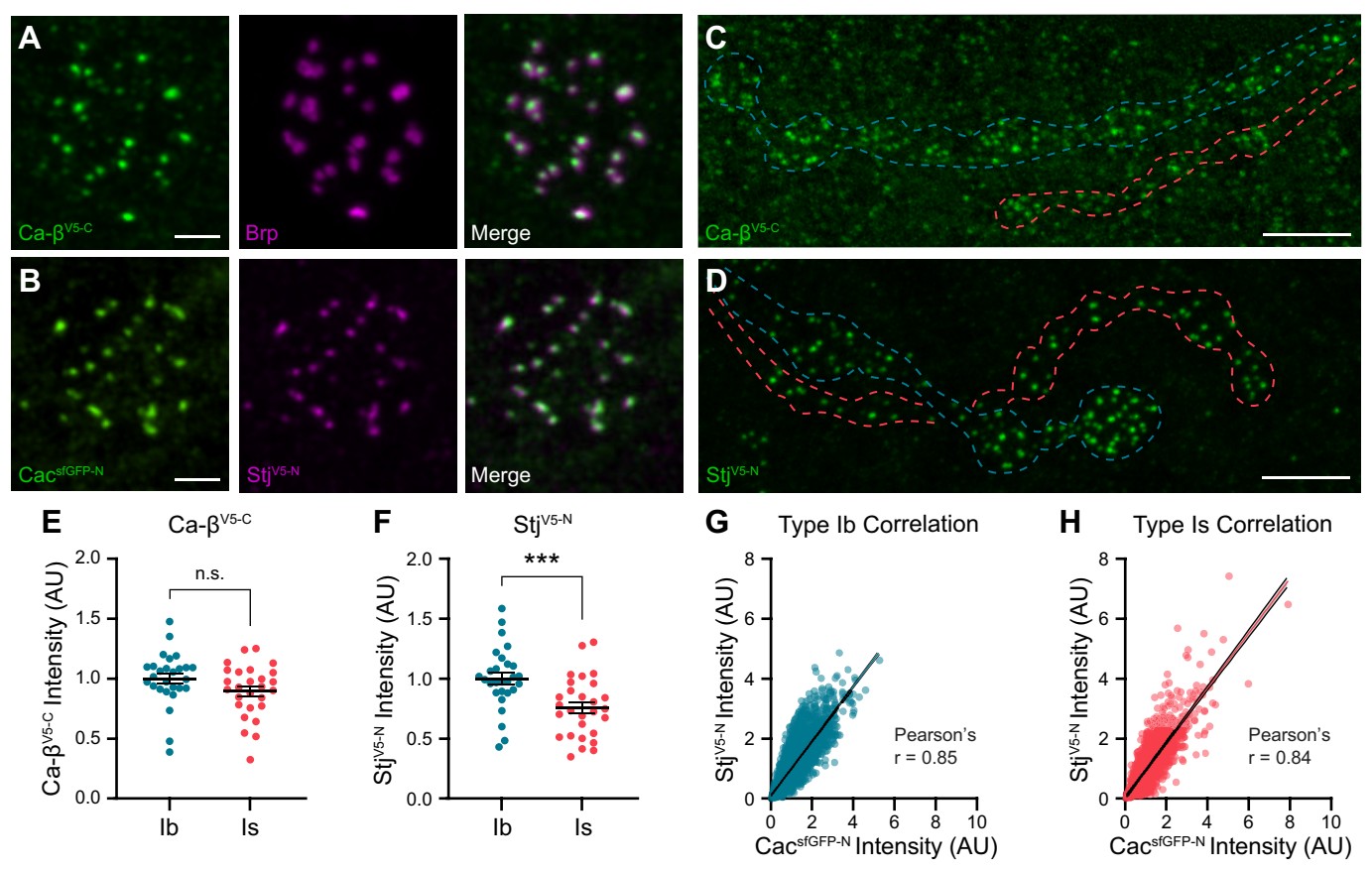

**Figure 6.** Stj/α2δ–3 levels are lower at AZs of high-$P_r$ type Is inputs. (**A**) Representative SoRa Z-projections of Ca-β$^{V5-C}$ (green), Brp (magenta), and merge at a single bouton. (**B**) Representative SoRa Z-projections of Cac$^{sfGFP-N}$ (green), Stj$^{V5-N}$ (magenta), and merge at a single bouton. Scale bars for A and B=1 µm. (**C, D**) Representative confocal Z-projections of Ca-β$^{V5-C}$ expression and Stj$^{V5-N}$ expression at type Ib (blue outline) and type Is (red outline) terminals. Scale bars = 5 µm. (**E, F**) Quantification of Ca-β$^{V5-C}$ and Stj$^{V5-N}$ fluorescence intensity at type Ib and Is AZs. Each data point represents the average normalized single AZ sum intensity for an individual NMJ. (**G, H**) Correlation of Cac$^{sfGFP-N}$ and Stj$^{V5-N}$ fluorescence intensity levels at type Ib and Is single AZs with linear regression lines (blue or red line, respectively) and 95% confidence intervals (black lines). All error bars indicate S.E.M. ***p<0.001; ns, not significant. N's, absolute values, and statistical information is detailed in ***Supplementary file 1a***.

leading to a pool of VGCCs lacking α2δ (***Müller et al., 2010***; ***Voigt et al., 2016***). Our results indicate this pool may be present in vivo and larger at high-$P_r$ type Is inputs.

To further investigate the contribution of Stj to synaptic heterogeneity, we analyzed the relationship between Cac and Stj levels at individual AZs of type Is inputs. Stj$^{V5-N}$ and Cac$^{sfGFP-N}$ levels are highly positively correlated at type Is AZs (***Figure 6G***). We observe the same relationship between Stj$^{V5-N}$ and Cac$^{sfGFP-N}$ levels at type Ib AZs (***Figure 6H***). Because $P_r$ is highly positively correlated with Cac levels within synaptic subtypes, this indicates that Stj levels are also positively correlated with $P_r$ within, but not between, inputs.

## Stj/α2δ–3 levels are modulated at AZs of both low- and high-$P_r$ inputs during presynaptic homeostatic potentiation

α2δ subunits are critical regulators of α subunit forward trafficking. In flies and mammals, overexpression of α2δ subunits increases α subunit abundance, whereas overexpression of the α subunit alone does not (***Cao et al., 2004***; ***Cunningham et al., 2022***; ***Hoppa et al., 2012***). These findings suggest that α2δ may be dynamically regulated together with Cac during PHP, a prediction we can now test with our endogenously tagged line. Following PhTx exposure, we find that Stj$^{V5-N}$ is recruited on a rapid timescale to both low- and high-$P_r$ AZs, increasing by a similar percentage at both type Ib and Is AZs (27% and 26%, respectively) as predicted (***Figure 7A-C***). Cac levels are similarly increased (33% at type Ib and 30% at type Is; See ***Figure 4D***), suggesting coordinated regulation. We have previously

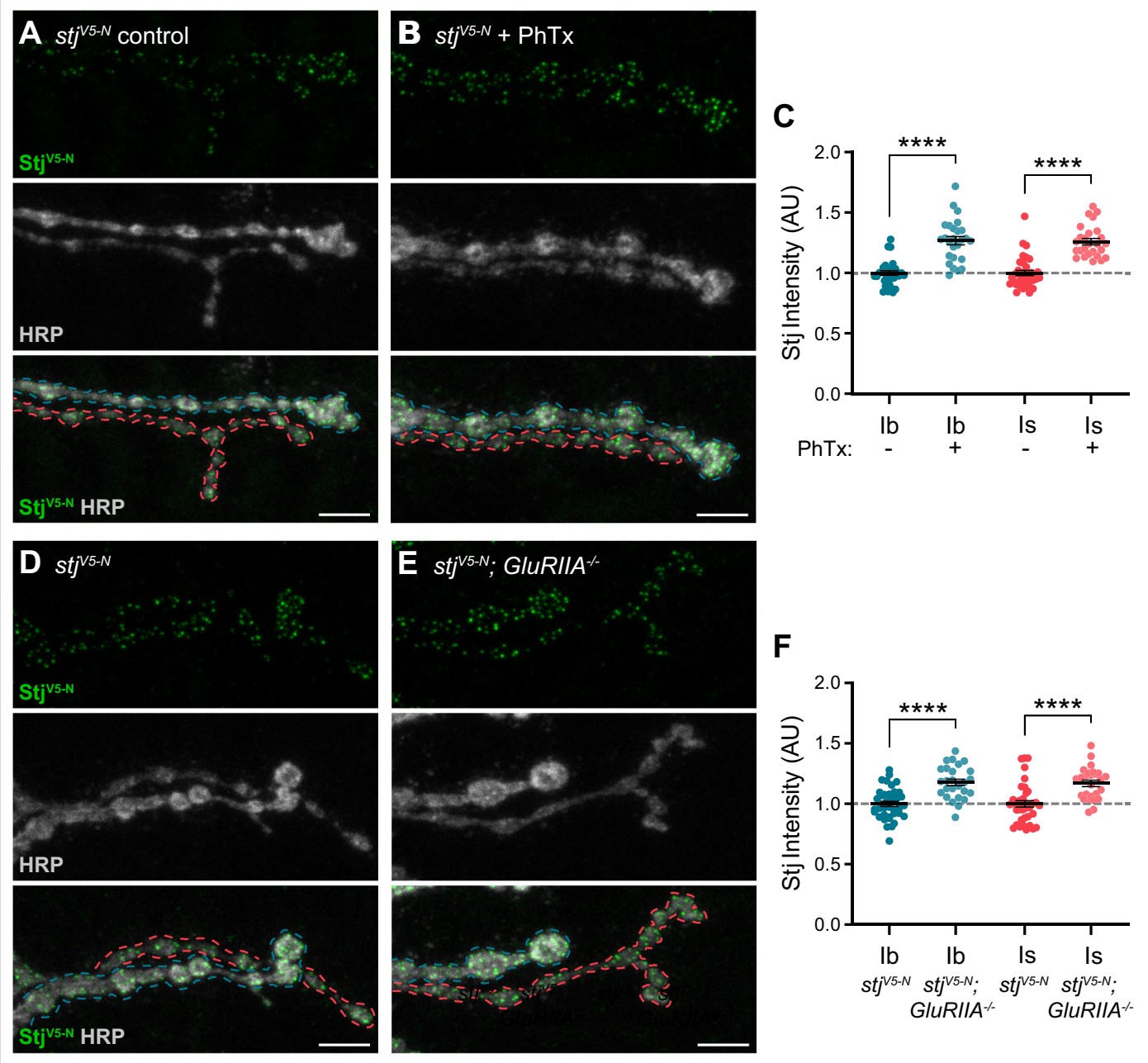

**Figure 7.** Stj/α2δ–3 levels are modulated at AZs of both low- and high-P$_r$ inputs during presynaptic homeostatic potentiation. (**A, B**) Representative confocal Z-projections of Stj$^{V5-N}$ (top, green), HRP (middle, gray), and merge (bottom) at untreated and PhTx-treated *stj*$^{V5-N}$ NMJs showing type Ib (blue) and type Is (red) terminals. (**C**) Quantification of Stj$^{V5-N}$ fluorescence intensity at untreated and PhTx-treated type Ib and Is terminals. (**D–E**) Representative confocal Z-projections of Stj$^{V5-N}$ (top, green), HRP (middle, gray), and merge (bottom) at *stj*$^{V5-N}$ and *stj*$^{V5-N}$;*GluRIIA*$^{-/-}$ NMJs showing type Ib (blue) and type Is (red) terminals. (**F**) Quantification of Stj$^{V5-N}$ fluorescence intensity levels at *stj*$^{V5-N}$ and *stj*$^{V5-N}$;*GluRIIA*$^{-/-}$ type Ib and Is terminals. For all quantifications, each data point represents the average normalized single AZ sum intensity for an individual NMJ. All scale bars = 5 µm, all error bars indicate S.E.M. ****p<0.0001. N's, absolute values, and statistical information is detailed in ***Supplementary file 1a***.

shown that Cac abundance is also increased in chronic PHP, which is induced by genetic loss of the GluRIIA receptor subunit (***Gratz et al., 2019***; ***Li et al., 2018***; ***Petersen et al., 1997***). We investigated Stj dynamics in *GluRIIA*$^{-/-}$ null animals and observed elevated Stj$^{V5-N}$ levels at AZs of both type Ib and Is inputs (18% and 17%, respectively; ***Figure 7D–F***), indicating similar dynamics during chronic PHP. These data are consistent with the recent finding that in addition to its previously uncovered role in promoting an increase in the readily releasable pool of SVs (***Wang et al., 2016***), Stj is required for the

accumulation of Cac at type Ib AZs during both acute and chronic PHP (*Zhang et al., 2023*). Thus, the abundance of multiple key AZ proteins distinguishes low- and high-$P_r$ synapses within, but not between, inputs at baseline and during homeostatic plasticity.

## Discussion

Complex nervous system function depends on communication at synapses with heterogeneous and plastic properties. Paradoxical findings in the field have raised questions about the role of VGCCs in establishing neurotransmitter release properties. Our findings suggest a model in which two broad intersecting mechanisms contribute to synaptic diversity in the nervous system: (1) nanoscale spatial organization and relative molecular content establish distinct average basal release probabilities that differ between inputs and (2) coordinated modulation of VGCC and active zone protein abundance independently tunes $P_r$ among individual synapses of distinct inputs. This model provides a framework for integrating diverse findings in the field and understanding how multiple levels of molecular and organizational diversity can intersect to generate extensive synaptic heterogeneity. Investigations at diverse synapses using approaches ranging from cell-attached patch recordings to freeze-fracture immuno-electron microscopy to correlative functional imaging have revealed a strong positive correlation between VGCC number and $P_r$ (*Akbergenova et al., 2018*; *Gratz et al., 2019*; *Holderith et al., 2012*; *Miki et al., 2017*; *Nakamura et al., 2015*; *Newman et al., 2022*; *Sheng et al., 2012*). This holds true among mature synapses in the hippocampus or immature synapses of the calyx of Held (*Holderith et al., 2012*; *Sheng et al., 2012*) and at the developing *Drosophila* NMJ where the differences in $P_r$ and channel number between AZs of a single input also correlate with synapse maturity (*Akbergenova et al., 2018*; *Gratz et al., 2019*; *Newman et al., 2022*). While this conclusion corresponds neatly with the dependence of neurotransmitter release on $Ca^{2+}$ influx, counterintuitively, a disconnect between VGCC and $P_r$ is observed in some studies. Although the number of functional VGCCs positively correlates with $P_r$ among synapses in the immature calyx of Held, the mature calyx has higher $P_r$ yet recruits fewer $Ca^{2+}$ channels (*Fedchyshyn and Wang, 2005*; *Sheng et al., 2012*; *Wang and Augustine, 2014*). Similarly, in the cerebellum, inhibitory stellate cells form high-$P_r$ synapses with lower VGCC levels than low-$P_r$ synapses formed by excitatory granular cells (*Rebola et al., 2019*). Adding to these paradoxical examples, we find that VGCC levels positively correlate with $P_r$ among the heterogeneous synapses formed by either low-$P_r$ type Ib or high-$P_r$ type Is motor neurons, but overall VGCC abundance is similar at the two inputs (*Figures 1 and 2*) despite an ~threefold difference in $P_r$ (*Lu et al., 2016*; *Newman et al., 2017*). Accordingly, our correlative functional imaging confirms that the same number of channels can support greater release at these high-$P_r$ AZs (*Figure 1*). While two-fold greater $Ca^{2+}$ influx at high-$P_r$ type Is AZs can explain much of this difference (*He et al., 2023*; *Lu et al., 2016*), mounting evidence suggests that here and in other systems differences in $P_r$ are separable from $Ca^{2+}$ influx. At distinct inputs formed by CA1 pyramidal neurons, $Ca^{2+}$ influx is greater at high-$P_r$ synapses, but doesn't explain differences in synaptic strength as raising $Ca^{2+}$ entry at low-$P_r$ synapses to high-$P_r$ synapse levels was not sufficient to increase synaptic strength to high-$P_r$ input levels (*Aldahabi et al., 2022*). Similar findings have been reported at tonic and phasic synapses of the Crayfish NMJ (*Msghina et al., 1999*).

The separability of VGCC abundance, $Ca^{2+}$ influx, and $P_r$ appears to be due to molecular and spatial differences between synaptic subtypes. In CA1 pyramidal neurons, differences in Munc-13-dependent SV priming are proposed to establish synapse-specific release properties, possibly due to the presence of distinct isoforms at low- vs. high-$P_r$ connections. In the cerebellum, fewer VGCC are more tightly coupled to SVs at high-$P_r$ stellate synapses (*Rebola et al., 2019*). More densely organized VGCCs at the mature vs. developing calyx of Held also exhibit greater coupling with SVs (*Chen et al., 2015*; *Fedchyshyn and Wang, 2005*; *Fekete et al., 2019*; *Nakamura et al., 2015*; *Sheng et al., 2012*). We find that Cac clusters are denser at high-$P_r$ AZs formed by *Drosophila* phasic motor neurons (*Figure 2*). Brp, which organizes both VGCCs and SVs, is also more densely organized at type Is synapses (*He et al., 2023*; *Jetti et al., 2023*; *Mrestani et al., 2021*), consistent with an overall more compact organization of high-$P_r$ AZs. A straightforward prediction is that a more compact AZ organization will decrease the distance between VGCCs and SVs. Indeed, a recent electrophysiological study using new tools for genetically isolating type Ib and Is inputs demonstrated that neurotransmitter release at denser type Is synapses is less impacted by the slow $Ca^{2+}$ chelator EGTA than type Ib synapses, indicating tighter VGCC-SV coupling (*He et al., 2023*). Together, these findings suggest

that more compact AZs may be a common organizing principle of high-$P_r$ synapses. Consistent with this model, Brp and Unc-13 density increase during PHP when $P_r$ is increased (*Dannhäuser et al., 2022*; *Ghelani et al., 2023*; *Mrestani et al., 2021*).

We also observe molecular differences between *Drosophila* motor inputs that may contribute to distinct $P_r$. Somewhat counterintuitively, high-$P_r$ type Is AZs have lower levels of both Brp and Stj/α2δ–3 (*Figures 3 and 6*). Brp/CAST/ELKS AZ cytomatrix proteins are central regulators of synapse organization across species (*Dai et al., 2006*; *Dong et al., 2018*; *Hallermann et al., 2010*; *Held et al., 2016*; *Kittel et al., 2006*; *Liu et al., 2014*; *McDonald et al., 2020*; *Radulovic et al., 2020*). Brp interacts broadly with AZ proteins to promote Cac clustering, organize SVs, and recruit Unc13A, which defines SV release sites (*Böhme et al., 2016*; *Fulterer et al., 2018*; *Ghelani et al., 2023*; *Liu et al., 2011*). While Brp clearly plays a central role in AZ organization and reorganization, we also find that type Ib and Is synapses have distinct requirements for Brp during synapse formation and homeostatic potentiation (*Figures 3 and 4*). Synapse-specific roles for Brp are supported by a recent functional imaging study in *Drosophila* (*Jetti et al., 2023*) and studies of ELKS at mammalian inhibitory and excitatory synapses (*Held et al., 2016*), and suggest additional factors establish neuron-specific differences in Brp dependence. A recent single-cell transcriptomic study of type Ib and Is motor neurons provides an unbiased starting point for identifying candidate regulators of molecular differences at low- and high-$P_r$ AZs (*Jetti et al., 2023*). Differentially expressed genes encode cytoskeletal and motor-related proteins, regulators of proteostasis, and post-translational modifying enzymes/pathway components – all of which could potentially contribute to establishing the observed molecular and/or spatial differences between type Ib and Is AZs. The VGCC complex itself provides an additional potential mechanism for diversifying synaptic function. Both β and α2δ subunits can influence the membrane localization and function of VGCCs (*Campiglio and Flucher, 2015*; *Dolphin and Lee, 2020*; *Weiss and Zamponi, 2017*). In addition to the ability to mix and match subunits, many of the genes encoding VGCC subunits across species are extensively alternatively spliced to generate additional functional diversity (*Cingolani et al., 2023*; *Lipscombe et al., 2013*; *Lipscombe and Lopez Soto, 2019*) – an area of great interest for further investigation at the *Drosophila* NMJ. Because both β and α2δ subunits are generally considered positive regulators of channel trafficking and function, we were surprised to find upon endogenously tagging Stj/α2δ–3 that its levels are lower at high-$P_r$ AZs (*Figures 5 and 6*). Since AZ levels of α subunit Cac are similar at the two inputs, lower levels of Stj at type Is synapses indicates a difference in α:α2δ–3 stoichiometry between low- and high-$P_r$ synaptic subtypes. While some previous studies have observed a tight association between the two subunits, a single-molecule tracking and modeling study of mammalian VGCCs found that α2δ subunits have a relatively low affinity for α subunits and predicted a population of α subunits not associated with an α2δ subunit (*Cassidy et al., 2014*; *Voigt et al., 2016*). Consistently, whereas α and β subunits were isolated at near equimolar ratios following affinity purification of Ca$_v$2 channels, molar levels of α2δ were a surprising 90% lower (*Müller et al., 2010*). Our findings suggest there may be a pool of VGCCs lacking an α2δ subunit at endogenous synapses, and further suggest that this pool is specific to or enriched at high-$P_r$ type Is AZs. Stj is both required (*Dickman et al., 2008*; *Kurshan et al., 2009*; *Ly et al., 2008*) and rate limiting (*Cunningham et al., 2022*) for Cac accumulation at AZs. Stj does not appear to function in the stabilization of channels at the AZ membrane, but rather at an upstream step in the progression from ER to plasma membrane (*Cunningham et al., 2022*), so complexes may not need to be maintained. However, a recent study in *C. elegans* suggested that auxiliary subunits to promote stabilization in the membrane, raising the possibility that the difference observed in α2δ levels could translate to a difference in α subunit mobility within Ib vs. Is AZs (*Oh et al., 2023*). Tools for following Stj dynamics in developing neurons will help clarify its precise role in Cac delivery at type Ib and Is AZs.

How might a higher α:α2δ–3 ratio result in higher $P_r$? One possibility involves α2δ–3 interactions with cell adhesion molecules. In mammals, α-Neurexin specifically inhibits Ca$^{2+}$ currents in Ca$_v$2.2 channels containing α2δ–3 (*Tong et al., 2017*), which, if similar at the *Drosophila* NMJ, could result in greater inhibition of channel function at type Ib AZs and contribute to the observed difference in Ca$^{2+}$ influx. Loss of *Drosophila* Neurexin reduces neurotransmitter release at the NMJ, but it also leads to reduced synaptogenesis so whether this is a direct effect remains unknown (*He et al., 2007*). Another possibility is that Stj isoforms with different functions are differentially expressed at type Ib and Is AZs. A recent transcriptomic study observed no significant differences in mRNA splice isoforms,

but transcript levels do not always reflect protein levels (*Jetti et al., 2023*). Notably, we find that among synapses of either low- or high-$P_r$ AZs, Stj levels correlate with $P_r$ and Stj levels are increased across AZs of both inputs during acute and chronic PHP (*Figure 7*). Cac and Stj levels increase by similar amounts at the two inputs, suggesting the difference in stoichiometry is maintained following homeostatic potentiation. α2δ–3 is an important drug target for treating epilepsy, neuropathic pain, and anxiety, so understanding its cell-specific roles and how α:α2δ–3 stoichiometry impacts channel function and contributes to synaptic diversity is of great interest.

## Materials and methods
### *Drosophila* genetics and gene editing
The following fly lines used in this study were obtained from the Bloomington *Drosophila* Stock Center (BDSC, NIH P40OD018537): *w1118* (RRID:BDSC_5905), *vasa-Cas9* (RRID:BDSC_51324), piggyBac transposase (RRID:BDSC_8283), and *Df(2 R)brp6.1* (*Gratz et al., 2014*; *Horn et al., 2003*). *brp69* and *GluRI-IAsp16* (*GluRIIA-/-*) alleles were generously provided by Stephan Sigrist (Freie Universität Berlin; *Fouquet et al., 2009*; *Kittel et al., 2006*) and Aaron DiAntonio (*Petersen et al., 1997*) respectively. *brp* loss-of-function experiments were performed in *brp69/Df(2 R)brp6.1*. *Drosophila melanogaster* stocks were raised on molasses food (Lab Express, Type R) in a 25 °C incubator with controlled humidity and 12 hr light/dark cycle. Endogenously tagged *cac*, *Ca-β*, *stolid*, and *straightjacket* (*stj*) alleles were generated using our piggyBac-based CRISPR approach as previously detailed (https://flycrispr.org/; *Bruckner et al., 2017*; *Gratz et al., 2019*). We used FlyBase (release FB2024_01) to obtain genomic sequences (doi.org/10.1093/genetics/iyad211). Td-Tomato and Halo tags were incorporated at the N-terminus of Cac, where we have previously incorporated fluorescent tags (*Ghelani et al., 2023*; *Gratz et al., 2019*). V5 tags were incorporated at the N-terminus of Stj and Stolid after their signal peptide sequences, immediately after Asparagine 34 of Stj and after Isoleucine 34 of Stolid. For Ca-β, V5 was inserted immediately after Proline 446 of isoform Ca-β-PN. All endogenously tagged lines are fully viable in homozygous males and females and were molecularly confirmed by Sanger sequencing. All genetic reagents generated in this study are available upon request. gRNAs sequences used:

>*cacHaloTag-N* / *cacTd-Tomato-N*: CATCGCTTAGCTGATAGAATGG
>*stjV5-N*: GCTGGCTGCAGATTGACGCACGG
>*stolidV5-N*: ATTTGTTGCATTCGCCGATCAGG
>*Ca-β V5-C*: CTCCGCAGATCCCGCGCGTCTGG

### Immunostaining
All antibodies used, associated fixation methods, and incubation times can be found in *Supplementary file 1b*. Male wandering third-instar larvae were dissected in ice-cold saline and fixed either for 6 min at room temperature with Bouin's fixative, 5 min on ice with 100% methanol, or 30 min at room temperature (RT) in 4% PFA. Dissections were permeabilized with PTX (PBS with 0.1% Triton-X 100) and blocked for 1 hr at RT using 5% goat serum and 1% bovine serum albumin. Stained larvae were mounted in Vectashield (Vector Laboratories, #H-1000) under Fisherbrand coverglass (Fisher Scientific, #12541B) for confocal microscopy, with Prolong glass mounting medium (Thermo Fisher Scientific, #P36980) under Zeiss High Performance Coverglass (Zeiss, #474030-9000-000) for super-resolution optical reassignment microscopy, or buffer (see STORM imaging and analysis section) under Zeiss coverglass with edges sealed using vacuum grease for STORM microscopy.

### Ca²⁺ imaging and analysis
Functional imaging was performed on a Nikon A1R resonant scanning confocal mounted on a FN1 microscope using a Nikon Apo LWD 25x1.1 NA objective and a Mad City Labs piezo drive nosepiece. Dissections and data collection were performed as previously described in *Gratz et al., 2019*. Briefly, *cacTd-Tomato-N*; *Mhc-GCaMP6f* male 3rd instar larvae were dissected in HL3 containing 0.2 mM Ca²⁺ and 25 mM Mg²⁺ with motor axons severed and the larval brain removed. Larval filets were placed in HL3 containing 1.5 mM Ca²⁺ and 25 mM Mg²⁺ for recording. Nerves were suctioned into 1.5 mm pipettes and stimulus amplitude was adjusted to recruit both type Ib and Is inputs. Motor terminals from segments A2-4 at NMJ 6/7 were imaged for CacTd-Tomato-N levels first using a galvanometer scanner,

then a resonant scanner to collect GCaMP6f events in a single focal plane continuously for 120 stimulations at a 0.2 Hz stimulation frequency.

Z-stacks and movies were loaded into Nikon Elements Software (NIS) where movies were motion corrected, background subtracted, and denoised. Change in fluorescence ($\Delta F$) movies were then created by subtracting the average of the previous 10 frames from each frame. A substack of only stimulation frames was further processed using a gaussian filter followed by the Bright Spots detection module in the Nikon GA3 software to identify the location of each postsynaptic event. Cac[Td-Tomato-N] fluorescence intensity levels and coordinate locations were measured for 531 AZs for type Ib and 365 AZs for type Is terminals across six animals. X-Y coordinate positions of fluorescent signals from GCaMP6f postsynaptic events were aligned to Cac[Td-Tomato-N] puncta locations and each post synaptic event assigned to a Cac punctum using nearest neighbor analysis. Postsynaptic events that did not map within 960 nm of a Cac[Td-Tomato-N] punctum were discarded from the analysis. Pearson's correlation was used to determine the correlation between $P_r$ and Cac levels normalized to average to account for variability between imaging sessions. Cac intensity-$P_r$ heat maps were generated using Python matplotlib and seaborn plotting packages.

## STORM imaging and analysis

STORM imaging was performed on a Nikon Eclipse Ti2 3D NSTORM with an Andor iXon Ultra camera, Nikon LUN-F 405/488/640 nm lasers, and a Nikon 100x1.49 NA objective. STORM buffer (10 mM MEA (pH 8.0), 3 U/mL pyranose oxidase, and 90 U/mL catalase, 10% (w/v) glucose, 10 mM sodium chloride, and 50 mM Tris hydrochloride) was made fresh each imaging day and pH adjusted to between 7.0–8.0 using acetic acid. *cac[HaloTag-N]* NMJs were labeled as detailed in *Supplementary file 1b* and immediately imaged for HRP in 488 channel to identify type Ib and Is terminals. Cac[HaloTag-N] was then imaged using the 640 nm laser line at 33 Hz for 5000 frames. 405 nm laser power was gradually increased over the course of imaging to compensate for run-down of blinking rates. A back aperture camera was used to ensure beam focus and position for each imaging session to ensure high signal to noise. Data were binned with a CCD minimum threshold of 100 and drift correction was applied using the NIS Software STORM package. ROIs of single boutons were drawn in NIS using HRP in the 488 nm channel followed by a DBSCAN analysis with criteria of 10 molecules within 50 nm to determine clusters. Positional coordinates of localizations within clusters from DBSCAN were exported from NIS and run through a Python script published with this manuscript. Using the implementation developed in *Mrestani et al., 2021* as a starting point, we wrote custom code to use the alpha shapes component of the CGAL package (https://www.cgal.org), via a python wrapper (https://anaconda.org/conda-forge/cgal), to measure the area of Ca$^{2+}$ channel clusters, the number of localizations, and calculate cluster density. To achieve an average lateral localization accuracy of ~30 nm, all localizations with >50 nm localization accuracy were removed prior to analysis. Using this custom code, Cac[HaloTag-N] area was analyzed using an alpha value of 0.015, which controls the complexity of cluster boundaries (not restricted to be convex).

## Confocal imaging and analysis

For quantitative AZ analysis of larval NMJs, dissections stained in the same dish were imaged on a Nikon Eclipse Ni A1*R*+confocal microscope using an Apo TIRF 60x1.49 NA oil-immersion objective for larval NMJs. NMJs containing both type Is and type Ib branches from muscles 6/7 in segments A2-4 were collected. ROIs were drawn using HRP staining to differentiate between type Ib and Is branches. To analyze individual AZs, Nikon Elements GA3 Software was used to process images with Gaussian and rolling ball filters and measure fluorescence intensity levels at individual puncta identified by the Bright Spots module. When experimental design allowed, Brp fluorescence signal was used to create a binary mask to aid in the identification of AZ ROIs for analysis. Otherwise, binary masks were created based on the fluorescence signal of the channel analyzed. Quantifications were conducted masked to genotype and/or treatment. Confocal fluorescence intensity level data are reported as the sum fluorescence intensity per AZ averaged over individual NMJs. For *Figure 5*, larvae were stained separately and imaged using a Nikon Plan-Apo 20x0.75 NA objective (ventral ganglia) or Apo TIRF 60x1.49 NA oil-immersion objective (NMJs). Super-resolution optical reassignment images were obtained on a Nikon CSU-W1 SoRa (Spinning Disk Super Resolution by Optical Pixel Reassignment)

with a Photometrics Prime BSI sCMOS camera and a 60x1.49 NA oil-immersion objective. Images were acquired using Nikon NIS and deconvolved using Richardson-Lucy deconvolution with 15–20 iterations.

## Electrophysiology

Current-clamp recordings were performed as previously described (*Bruckner et al., 2017*). Male third-instar larvae were dissected in HL3 (70 mM NaCl, 5 mM KCl, 15 mM MgCl2, 10 mM NaHCO3, 115 mM sucrose, 5 mM trehalose, 5 mM HEPES, pH 7.2) with 0.25 mM $Ca^{2+}$. Recordings were performed in HL3 at the external $Ca^{2+}$ concentration indicated. Sharp borosilicate electrodes filled with 3 M KCl were used to record from muscle 6 of abdominal segments A3 and A4. Recordings were conducted on a Nikon FN1 microscope using a 40x0.80 NA water-dipping objective and acquired using an Axoclamp 900 A amplifier, Digidata 1550B acquisition system, and pClamp 11.0.3 software (Molecular Devices). For each cell with an initial resting potential between −60 and −80 mV and input resistance ≥5 MΩ, mean miniature excitatory junctional potentials (mEJPs) were collected for 1 min in the absence of stimulation and analyzed using Mini Analysis (Synaptosoft). EJPs were generated by applying a stimulus to severed segmental nerves at a frequency of 0.2 Hz using an isolated pulse stimulator 2100 (A-M Systems). Stimulus amplitude was adjusted to consistently elicit compound responses from both type Ib and Is motor neurons. At least 25 consecutive EJPs were recorded for each cell and analyzed in pClamp to obtain mean amplitude. Quantal content was calculated for each recording as mean EJP amplitude divided by mean mEJP amplitude.

## Acute homeostatic challenge

Acute PHP was induced by incubating semi-intact preparations in 20 µM Philanthotoxin-433 (*Figure 4*: PhTx; Santa Cruz, sc-255421, Lot B1417 and *Figure 7*: PhTx; Sigma Aldrich, P207-2, Lot MKCK7405) diluted in HL3 containing 0.4 mM $Ca^{2+}$ for 10 min at room temperature (*Frank et al., 2006*). Control preparations were given a mock treatment. Following control and experimental treatment, dissections were completed, fixed in 4% PFA for 30 min (*cac*$^{sfGFP-N}$) or 100% ice-cold methanol on ice for 5 min (*stj*$^{V5-N}$), and stained in the same dish. Analyses of fluorescent intensity levels were performed as previously described in the *Confocal imaging and analysis section*.

## Experimental design and statistical analysis

Statistical analyses were conducted in GraphPad Prism 9. Normality was determined by the D'Agostino–Pearson omnibus test. Comparisons of normally distributed data were conducted by Student's *t* test (with Welch's correction in the case of unequal variance) for single comparisons and ANOVA followed by Tukey's test for multiple comparisons. For non-normally distributed data, the Mann–Whitney *U* test and Kruskal-Wallis test followed by Dunn's multiple comparisons tests were used for single and multiple comparisons, respectively. Paired analysis of non-normally distributed data was conducted using Wilcoxon's matched-pairs signed rank test. One-dimensional Pearson correlation coefficients (*r*) were used to compare intensity levels and neurotransmitter release probability. ANCOVA test was performed on all regression lines to determine if slopes were significantly different. Reported values are mean ± SEM. Sample size, statistical test, and *p* values for each comparison are reported in *Supplementary file 1*a. All source data, including statistical tests and raw images, can be found in the following Harvard Dataverse dataset https://doi.org/10.7910/DVN/GGP3UM.

## Acknowledgements

We thank the Developmental Studies Hybridoma Bank, the Bloomington Drosophila Stock Center (NIH P40OD018537), Flybase (*Öztürk-Çolak et al., 2024*), Ehud Isacoff (UC Berkeley), and Stephan Sigrist (Freie Universität Berlin) for providing antibodies and fly stocks. The Nikon SoRa/STORM microscope was generously provided by The Neurobiology of Cells and Circuits/Center for Translational Neuroscience Microscopy Committee, Carney Institute for Brain Science. We are grateful to Joel Hirsch (Tel Aviv University) for consultations on tagging Ca-β, Nicholas Deakin (Nikon) for guidance on STORM imaging, the Heckmann lab (University of Würzburg) for guidance on STORM image analysis pipelines, Matthew Knoeppel for help generating CRISPR alleles, and Liana Lewis for her assistance with image analysis. We thank Rajan Thakur and the members of the O'Connor-Giles lab for thoughtful discussions and comments on the manuscript. This work was supported by grants from the National

Institute of Neurological Disorders and Stroke, National Institutes of Health to KMOG (R01NS078179) and ATM (F31NS122424), Brown Neuroscience Graduate Program training grant T32 MH020068, and funds from the Brown University Carney Institute for Brain Science.

## Additional information

### Funding

| Funder | Grant reference number | Author |
| --- | --- | --- |
| National Institute of Neurological Disorders and Stroke | R01NS078179 | Kate M O'Connor-Giles |
| National Institute of Neurological Disorders and Stroke | F31NS122424 | Audrey T Medeiros |
| Brown Neuroscience Graduate Program | T32 MH020068 | Audrey T Medeiros |

The funders had no role in study design, data collection, and interpretation, or the decision to submit the work for publication.

### Author contributions

Audrey T Medeiros, Conceptualization, Resources, Data curation, Software, Formal analysis, Funding acquisition, Validation, Investigation, Visualization, Methodology, Writing – original draft, Project administration, Writing – review and editing; Scott J Gratz, Conceptualization, Resources, Data curation, Software, Formal analysis, Supervision, Funding acquisition, Validation, Investigation, Visualization, Methodology, Writing – original draft, Project administration, Writing – review and editing; Ambar Delgado, Data curation, Formal analysis, Investigation, Writing – review and editing; Jason T Ritt, Software, Formal analysis, Methodology, Writing – review and editing; Kate M O'Connor-Giles, Conceptualization, Data curation, Formal analysis, Supervision, Funding acquisition, Writing – original draft, Project administration, Writing – review and editing

### Author ORCIDs

Audrey T Medeiros ⓘ http://orcid.org/0000-0002-5562-4772
Scott J Gratz ⓘ http://orcid.org/0000-0002-0106-8336
Jason T Ritt ⓘ https://orcid.org/0000-0003-3113-7977
Kate M O'Connor-Giles ⓘ https://orcid.org/0000-0002-2259-8408

Reviewer #1 (Public Review): https://doi.org/10.7554/eLife.88412.3.sa1
Reviewer #2 (Public Review): https://doi.org/10.7554/eLife.88412.3.sa2
Author response https://doi.org/10.7554/eLife.88412.3.sa3

## Additional files

### Supplementary files

• Transparent reporting form

• Supplementary file 1. Tables detailing absolute values, statistics, and imaging conditions of endogenously-tagged genetic lines.

### Data availability

All source data, including statistical tests and raw images, can be found in the following Harvard Dataverse dataset https://doi.org/10.7910/DVN/GGP3UM.

The following dataset was generated:

| Author(s) | Year | Dataset title | Dataset URL | Database and Identifier |
| --- | --- | --- | --- | --- |
| Audrey M | 2024 | Source data for Medeiros et al., eLife Version of Record | https://doi.org/10.7910/DVN/GGP3UM | Harvard Dataverse, 10.7910/DVN/GGP3UM |

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
