## [Editor Report · eLife assessment]

Calcium channels are key regulators of synaptic strength and plasticity. The authors generate new endogenous tags of the *Drosophila* channel Cac as well as auxiliary subunits to investigate distinct calcium channel functions at the fly NMJ, Is and Ib. They demonstrate functions for voltage-gated calcium channel subunits in promoting synaptic strength, diversity, and plasticity with a series of **convincing** analyses. The work is **important** and has broad implications. In addition, the newly developed tools should be quite beneficial for fly biologists.

---

## [Referee Report · Reviewer #1 (Public Review)]

Calcium channels are key regulators of synaptic strength and plasticity, yet how these channels are differentially utilized to enable synaptic diversity is not clear. In this manuscript, the authors use new endogenous tagging of the *Drosophila* CaV2 channel Cac and three auxiliary subunits to investigate distinct calcium channel functions at two motor neuron subtypes at the fly NMJ, Is and Ib. Although it is clear from previous studies that Pr is higher at Is over Ib, it is not clear why. The authors confirm these differences using postsynaptic calcium imaging combined with post-hoc Cac-TdTomato imaging. Then, through a series of confocal and super resolution imaging studies, the authors describe differences in calcium channel and active zone structure between Is and Ib motor neuron terminals, and the role of Brp and homeostatic plasticity in regulating channel abundance. Finally, the authors show that while the CaBeta subunit is present at similar levels at Is and Ib active zones, there is an interesting reduction in Stj at Is active zones. The authors conclude that these differences in active zone structure and architecture contribute to the generation of the observed heterogeneity in synaptic strength.

Overall the manuscript is well written, and the successful generation of the new endogenous Cac tags (Td-Tomato, Halo) and CaBeta, stj, and stolid genes with V5 tags will be powerful reagents for the field to enable new studies on calcium channels in synaptic structure, function, and plasticity. There are also some interesting, though not entirely unexpected, findings regarding how Brp and homeostatic plasticity modulate calcium channel abundance. The key factors generating diversity in synaptic strength beyond simple Ca2+ influx are well articulated in framing this study. Beyond the particularly useful new reagents for the field presented, the new data demonstrating a concerted and coupled increase in Cac, Stj, and CaB together after plasticity provides an interesting new dimension to the study and a foundation for new work moving forward.

Comments on revision:

This is a much improved revised manuscript, where the authors have done an excellent job of responding to my initial concerns. In particular, the key factors generating diversity in synaptic strength beyond simple Ca2+ influx are better articulated in framing this study. Beyond the particularly useful new reagents for the field presented, the new data demonstrating a concerted and coupled increase in Cac, Stj, and CaB together after plasticity provides an interesting new dimension to the study and a foundation for new work moving forward.

Upon reflection, I think my initial review came across as a bit harsh, and I am happy to now update my original evaluation to better reflect the importance and impact of this very nice study. I commend the authors on an outstanding study.

---

## [Referee Report · Reviewer #2 (Public Review)]

The authors aim to investigate how voltage-gated calcium channel number, organization, and subunit composition lead to changes in synaptic activity at tonic and phasic motor neuron terminals, or type Is and Ib motor neurons in *Drosophila*. These neuron subtypes generate widely different physiological outputs, and many investigations have sought to understand the molecular underpinnings responsible for these differences. Additionally, these authors explore not only static differences that exist during the third-instar larval stage of development but also use a pharmacological approach to induce homeostatic plasticity to explore how these neuronal subtypes dynamically change the structural composition and organization of key synaptic proteins contributing to physiological plasticity. The *Drosophila* neuromuscular junction (NMJ) is glutamatergic, the main excitatory neurotransmitter in the human brain, so these findings not only expand our understanding of the molecular and physiological mechanisms responsible for differences in motor neuron subtype activity, but also contribute to our understanding of how the human brain and nervous system functions.

The authors employ state-of-the-art tools and techniques such as single-molecule localization microscopy 3D STORM and create several novel transgenic animals using CRISPR to expand the molecular tools available for exploration of synaptic biology that will be of wide interest to the field. Additionally, the authors use a robust set of experimental approaches from active zone level resolution functional imaging from live preparations to electrophysiology and immunohistochemical analyses to explore and test their hypotheses. All data appear to be robustly acquired and analyzed using appropriate methodology. The authors make important advancements to our understanding of how the different motor neuron subtypes, phasic and tonic-like, exhibit widely varying electrical output despite the neuromuscular junctions having similar ultrastructural composition in the proteins of interest, voltage gated calcium channel cacophony (cac) and the scaffold protein Bruchpilot (brp). The authors reveal the ratio of brp:cac appears to be a critical determinant of release probability (Pr), and in particular, the packing density of VGCCs and availability of brp. Importantly, the authors demonstrate a brp-dependent increase in VGCC density following acute philanthotoxin perfusion (glutamate receptor inhibitor). This VGCC increase appears to be largely responsible for the presynaptic homeostatic plasticity (PHP) observable at the Drosophila NMJ. Lastly, the authors created several novel CRISPR-tagged transgenic lines to visualize the spatial localization of VGCC subunits in *Drosophila*. Two of these lines, CaβV5-C and stjV5-N, express in motor neurons and in the nervous system, localize at the NMJ, and most strikingly, strongly correlate with Pr at tonic and phasic-like terminals.

---

## [Author Response]

The following is the authors’ response to the original reviews.

**Reviewer #1 (Public review):**
[...] Overall the manuscript is well written, and the successful generation of the new endogenous Cac tags (Td-Tomato, Halo) and CaBeta, stj, and stolid genes with V5 tags will be powerful reagents for the field to enable new studies on calcium channels in synaptic structure, function, and plasticity. There are also some interesting, though not entirely unexpected, findings regarding how Brp and homeostatic plasticity modulate calcium channel abundance. However, a major concern is that the conclusions about how "molecular and organization diversity generate functional synaptic heterogeneity" are not really supported by the data presented in this study. In particular, the key fact that frames this study is that Cac levels are similar at Ib and Is active zones, but that Pr is higher at Is over Ib (which was previously known). While Pr can be influenced by myriad processes, the authors should have first assessed presynaptic calcium influx - if they had, they would have better framed the key questions in this study. As the authors reference from previous studies, calcium influx is at least two-fold higher per active zone at Is over Ib, and the authors likely know that this difference is more than sufficient to explain the difference in Pr at Is over Ib. Hence, there is no reason to invoke differences in "molecular and organization diversity" to explain the difference in Pr, and the authors offer no data to support that the differences in active zone structure at Is vs Ib are necessary for the differences in Pr. Indeed, the real question the authors should have investigated is why there are such differences in presynaptic calcium influx at Is over Ib despite having similar levels/abundance of Cac. This seems the real question, and is all that is needed to explain the Pr differences shown in Fig. 1. The other changes in active zone structure and organization at Is vs Ib may very well contribute to additional differences in Pr, but the authors have not shown this in the present study, and rely on other studies (such as calcium-SV coupling at Is vs Ib) to support an argument that is not necessitated by their data. At the end of this manuscript, the authors have found an interesting possibility that Stj levels are reduced at Is vs Ib, that might perhaps contribute to the difference in calcium influx. However, at present this remains speculative.Overall, the authors have generated powerful reagents for the field to study calcium channels and how they are regulated, but draw conclusions about active zone structure and organization contributing to functional heterogeneity that are not strongly supported by the data presented.

Reviewer 1 raises an interesting question that we agree will form the basis of important studies. Here, we set out to address a different question, which we will work to better frame. While we and others had previously found a strong correlation between calcium channel abundance and synaptic release probability (Pr (Akbergenova et al., 2018; Gratz et al., 2019; Holderith et al., 2012; Nakamura et al., 2015; Sheng et al., 2012)), more recent studies found that calcium channel abundance does not necessarily predict synaptic strength (Aldahabi et al., 2022; Rebola et al., 2019). Our study explores this paradox and presents findings that provide an explanation: calcium channel abundance predicts Pr among individual synapses of either low-Pr type-Ib or high-Pr type-Is inputs where modulating channel number tunes synaptic strength, but does not predict Pr between the two inputs, indicating an inputspecific role for calcium channel abundance in promoting synaptic strength. Thus, we propose that calcium channel abundance predictably modulates synaptic strength among individual synapses of a single input or synapse subtype, which share similar molecular and spatial organization, but not between distinct inputs where the underlying organization of active zones differs. Consistently, in the mouse, calcium channel abundance correlates strongly with release probability specifically when assessed among homogeneous populations of connections (Aldahabi et al., 2022; Holderith et al., 2012; Nakamura et al., 2015; Rebola et al., 2019; Sheng et al., 2012).

As Reviewer 1 notes, the two-fold difference in calcium influx at type-Is synapses is certainly an important difference underlying three-fold higher Pr. However, growing evidence indicates that calcium influx alone, like calcium channel abundance, does not reliably predict synaptic strength between inputs. For example, Rebola et al. (2019) compared cerebellar synapses formed by granule and stellate cells and found that lower Pr granule synapses exhibit both higher calcium channel abundance and calcium influx. In another example, Aldahabi et al. (2023) demonstrate that even when calcium influx is greater at high-Pr synapses, it does not necessarily explain differences in synaptic strength between inputs. Studying excitatory hippocampal CA1 synapses onto distinct interneuronal targets, they found that raising calcium entry at low-Pr inputs to high-Pr synapse levels is not sufficient to increase synaptic strength to high-Pr synapse levels. Similarly, at the *Drosophila* NMJ, the finding that type-Ib synapses exhibit loose calcium channel-synaptic vesicle coupling whereas type-Is synapses exhibit tight coupling suggests factors beyond calcium influx also contribute to differences in Pr between the two inputs (He et al., 2023). Consistently, a two-fold increase in external calcium does not induce a three-fold increase in release at low-Pr type-Ib synapses (He et al., 2023). Thus, upon finding that calcium channel abundance is similar at type-Ib and -Is synapses, we focused on identifying differences beyond calcium channel abundance and calcium influx that might contribute their distinct synaptic strengths. We agree that these studies, ours included, cannot definitively determine the contribution of identified organizational differences to distinct release probabilities because it is not currently possible to specifically alter subsynaptic organization, and will ensure that our language is tempered accordingly. However, in addition to the studies cited above and our findings, recent work demonstrating that homeostatic potentiation of neurotransmitter release is accompanied by greater spatial compaction of multiple active zone proteins (Dannhauser et al., 2022; Mrestani et al., 2021) and decreased calcium channel mobility (Ghelani et al., 2023) provide support for the interpretation that subsynaptic organization is a key parameter for modulating Pr.

**Reviewer #2 (Public review):**
The authors aim to investigate how voltage-gated calcium channel number, organization, and subunit composition lead to changes in synaptic activity at tonic and phasic motor neuron terminals, or type Is and Ib motor neurons in *Drosophila*. These neuron subtypes generate widely different physiological outputs, and many investigations have sought to understand the molecular underpinnings responsible for these differences. Additionally, these authors explore not only static differences that exist during the third-instar larval stage of development but also use a pharmacological approach to induce homeostatic plasticity to explore how these neuronal subtypes dynamically change the structural composition and organization of key synaptic proteins contributing to physiological plasticity. The *Drosophila* neuromuscular junction (NMJ) is glutamatergic, the main excitatory neurotransmitter in the human brain, so these findings not only expand our understanding of the molecular and physiological mechanisms responsible for differences in motor neuron subtype activity but also contribute to our understanding of how the human brain and nervous system functions.The authors employ state-of-the-art tools and techniques such as single-molecule localization microscopy 3D STORM and create several novel transgenic animals using CRISPR to expand the molecular tools available for exploration of synaptic biology that will be of wide interest to the field. Additionally, the authors use a robust set of experimental approaches from active zone level resolution functional imaging from live preparations to electrophysiology and immunohistochemical analyses to explore and test their hypotheses. All data appear to be robustly acquired and analyzed using appropriate methodology. The authors make important advancements to our understanding of how the different motor neuron subtypes, phasic and tonic-like, exhibit widely varying electrical output despite the neuromuscular junctions having similar ultrastructural composition in the proteins of interest, voltage gated calcium channel cacophony (cac) and the scaffold protein Bruchpilot (brp). The authors reveal the ratio of brp:cac appears to be a critical determinant of release probability (Pr), and in particular, the packing density of VGCCs and availability of brp. Importantly, the authors demonstrate a brp-dependent increase in VGCC density following acute philanthotoxin perfusion (glutamate receptor inhibitor). This VGCC increase appears to be largely responsible for the presynaptic homeostatic plasticity (PHP) observable at the *Drosophila* NMJ. Lastly, the authors created several novel CRISPRtagged transgenic lines to visualize the spatial localization of VGCC subunits in *Drosophila*. Two of these lines, CaBV5-C and stjV5-N, express in motor neurons and in the nervous system, localize at the NMJ, and most strikingly, strongly correlate with Pr at tonic and phasic-like terminals.(1) The few limitations in this study could be addressed with some commentary, a few minor follow-up analyses, or experiments. The authors use a postsynaptically expressed calcium indicator (mhcGal4>UAS -GCaMP) to calculate Pr, yet do not explore the contribution that glutamate receptors, or other postsynaptic contributors (e.g. components of the postsynaptic density, PSD) may contribute. A previous publication exploring tonic vs phasic-like activity at the *Drosophila* NMJ revealed a dynamic role for GluRII (Aponte-Santiago et al, 2020). Could the speed of GluR accumulation account for differences between neuron subtypes?

We did observe that GCaMP signals are higher at type Is synapses, where synapses tend to form later but GluRs accumulate more rapidly upon innervation (Aponte-Santiago et al., 2020). However, because we are using our GCaMP indicator as a plus/minus readout of synaptic vesicle release at mature synapses, we do not expect differences in GluR accumulation to have a significant effect on our measures. Consistently, the difference in Pr we observe between type-Ib and -Is inputs (Fig. 1C) is similar to that previously reported (He et al., 2023; Lu et al., 2016; Newman et al., 2022).

(2) The observation that calcium channel density and brp:cac ratio as a critical determinant of Pr is an important one. However, it is surprising that this was not observed in previous investigations of cac intensity (of which there are many). Is this purely a technical limitation of other investigations, or are other possibilities feasible? Additionally, regarding VGCC-SV coupling, the authors conclude that this packing density increases their proximity to SVs and contributes to the steeper relationship between VGCCs and Pr at phasic type Is. Is it possible that brp or other AZ components could account for these differences. The authors possess the tools to address this directly by labeling vesicles with JanellaFluor646; a stronger signal should be present at Is boutons. Additionally, many different studies have used transmission electron microscopy to explore SVs location to AZs (t-bars) at the *Drosophila* NMJ.To date, the molecular underpinnings of heterogeneity in synaptic strength have primarily been investigated among individual type-Ib synapses. However, a recent study investigating differences between type-Ib and -Is synapses also found that the Cac:Brp ratio is higher at type-Is synapses (He et al., 2023).

At this point, we do not know which active zone components are responsible for the organizational (Figs. 1, 2) and coupling (now demonstrated by He et al., 2023) differences between type-Ib and -Is synapses or what establishes the differences in active zone protein levels we observe (Figs. 3,6), although Brp likely plays a local role. We find that Brp is required for dynamically regulating calcium channel levels during homeostatic plasticity and plays distinct roles at type-Ib and -Is synapses (Figs. 3, 4). Brp regulates a number of proteins critical for the distribution of docked synaptic vesicles near T bars of type Ib active zones, including Unc13 (Bohme et al., 2016). Extending these studies to type-Is synapses will be of great interest.

(3) In reference to the contradictory observations that VGCC intensity does not always correlate with, or determine Pr. Previous investigations have also observed other AZ proteins or interactors (e.g. synaptotagmin mutants) critically control release, even when the correlation between cac and release remains constant while Pr dramatically precipitates.

This is an important point as a number of molecular and organizational differences between high- and low-Pr synapses certainly contribute to baseline functional differences. The other proteins we (Figs. 3,6) and others (Dannhauser et al., 2022; Ehmann et al., 2014; He et al., 2023; Jetti et al., 2023; Mrestani et al., 2021; Newman et al., 2022) have investigated are less abundant and/or more densely organized at type-Is synapses. Investigating additional active zone proteins, including synaptic proteins, and determining how these factors combine to yield increased synaptic strength are important next steps.Thank you. We have corrected this error.

(4) To confirm the observations that lower brp levels results in a significantly higher cac:brp ratio at phasic-like synapses by organizing VGCCs; this argument could be made stronger by analyzing their existing data. By selecting a population of AZs in Ib boutons that endogenously express normal cac and lower brp levels, the Pr from these should be higher than those from within that population, but comparable to Is Pr. I believe the authors should also be able to correlate the cac:brp ratio with Pr from their data set generally; to determine if a strong correlation exists beyond their observation for cac correlation.

We do not have simultaneous measures of Pr and Cac and Brp abundance. However, our findings suggest that distinct Cac:Brp ratios at type Ib and Is inputs reflect underlying organizational differences that contribute to distinct release probabilities between the two synaptic subtypes. In contrast, within either synaptic subtype, release probability is positively correlated with both Cac and Brp levels. Thus, the mechanisms driving functional differences between synaptic subtypes are distinct from those driving functional heterogeneity within a subtype, so we do not expect Cac:Brp ratio to correlate with Pr among individual type-Ib synapses. We will work to clarify this point in the revised text.

(5) For the philanthotoxin induced changes in cac and brp localization underlying PHP, why do the authors not show cac accumulation after PhTx on live dissected preparations (i.e. in real time)? This also be an excellent opportunity to validate their brp:cac theory. Do the authors observe a dynamic change in brp:cac after 1, or 5 minutes; do Is boutons potentiate stronger due to proportional increases in cac and brp? Also regarding PhTx-induced PHP, their observations that stj and α2δ-3 are more abundant at Is synapses, suggests that they may also play a role in PhTx induced changes in cac. If either/both are overexpressed during PhTx, brp should increase while cac remains constant. These accessory proteins may determine cac incorporation at AZs.

As we have previously followed Cac accumulation in live dissected preparations and found that levels increase proportionally across individual synapses (Gratz et al., 2019), we did not attempt to repeat these challenging experiments at smaller type-Is synapses. We will reanalyze our data to investigate Cac:Brp ratio at individual active zones post PhTx. However, as noted above, we do not expect changes in the Cac:Brp ratio to correlate with Pr among individual synapses of single inputs as this measure reflects organization differences between inputs and PhTx induces an increase in the abundance of both proteins at both inputs.

Determining the effect of PhTx on Stj levels at type-Ib and -Is active zones is an excellent idea and might provide insight into how lower Stj levels correlate with higher Pr at type-Is synapses. While prior studies have demonstrated critical roles for Stj in regulating Cac accumulation during development and in promoting presynaptic homeostatic potentiation (Cunningham et al., 2022; Dickman et al., 2008; Kurshan et al., 2009; Ly et al., 2008; Wang et al., 2016), its regulation during PHP has not been investigated.

Taken together this study generates important data-driven, conceptional, and theoretical advancements in our understanding of the molecular underpinnings of different motor neurons, and our understanding of synaptic biology generally. The data are robust, thoroughly analyzed, appropriately depicted. This study not only generates novel findings but also generated novel molecular tools which will aid future investigations and investigators progress in this field.

References

Akbergenova, Y., K.L. Cunningham, Y.V. Zhang, S. Weiss, and J.T. Littleton. 2018. Characterization of developmental and molecular factors underlying release heterogeneity at *Drosophila* synapses. eLife. 7.

Aldahabi, M., F. Balint, N. Holderith, A. Lorincz, M. Reva, and Z. Nusser. 2022. Different priming states of synaptic vesicles underlie distinct release probabilities at hippocampal excitatory synapses. Neuron. 110:4144-4161 e4147.

Aponte-Santiago, N.A., K.G. Ormerod, Y. Akbergenova, and J.T. Littleton. 2020. Synaptic Plasticity Induced by Differential Manipulation of Tonic and Phasic Motoneurons in *Drosophila*. The Journal of neuroscience : the official journal of the Society for Neuroscience. 40:6270-6288.

Bohme, M.A., C. Beis, S. Reddy-Alla, E. Reynolds, M.M. Mampell, A.T. Grasskamp, J. Lutzkendorf, D.D. Bergeron, J.H. Driller, H. Babikir, F. Gottfert, I.M. Robinson, C.J. O'Kane, S.W. Hell, M.C. Wahl, U. Stelzl, B. Loll, A.M. Walter, and S.J. Sigrist. 2016. Active zone scaffolds differentially accumulate Unc13 isoforms to tune Ca(2+) channel-vesicle coupling. Nature neuroscience. 19:1311-1320.

Cunningham, K.L., C.W. Sauvola, S. Tavana, and J.T. Littleton. 2022. Regulation of presynaptic Ca(2+) channel abundance at active zones through a balance of delivery and turnover. Elife. 11.

Dannhauser, S., A. Mrestani, F. Gundelach, M. Pauli, F. Komma, P. Kollmannsberger, M. Sauer, M. Heckmann, and M.M. Paul. 2022. Endogenous tagging of Unc-13 reveals nanoscale reorganization at active zones during presynaptic homeostatic potentiation. Front Cell Neurosci. 16:1074304.

Dickman, D.K., P.T. Kurshan, and T.L. Schwarz. 2008. Mutations in a *Drosophila* alpha2delta voltage gated calcium channel subunit reveal a crucial synaptic function. The Journal of neuroscience : the official journal of the Society for Neuroscience. 28:31-38.

Ehmann, N., S. Van De Linde, A. Alon, D. Ljaschenko, X.Z. Keung, T. Holm, A. Rings, A. Diantonio, S. Hallermann, U. Ashery, M. Heckmann, M. Sauer, and R.J. Kittel. 2014. Quantitative super-resolution imaging of Bruchpilot distinguishes active zonestates. Nature Communications. 5.

Ghelani, T., M. Escher, U. Thomas, K. Esch, J. Lützkendorf, H. Depner, M. Maglione, P. Parutto, S. Gratz, T. Matkovic-Rachid, S. Ryglewski, A.M. Walter, D. Holcman, K. O‘Connor Giles, M. Heine, and S.J. Sigrist. 2023. Interactive nanocluster compaction of the ELKS scaffold and Cacophony Ca^2+^ channels drives sustained active zone potentiation. Science Advances. 9:eade7804.

Gratz, S.J., P. Goel, J.J. Bruckner, R.X. Hernandez, K. Khateeb, G.T. Macleod, D. Dickman, and K.M. O'Connor-Giles. 2019. Endogenous tagging reveals differential regulation of Ca^2+^ channels at single AZs during presynaptic homeostatic potentiation and depression. The Journal of Neuroscience:3068-3018.

He, K., Y. Han, X. Li, R.X. Hernandez, D.V. Riboul, T. Feghhi, K.A. Justs, O. Mahneva, S. Perry, G.T. Macleod, and D. Dickman. 2023. Physiologic and Nanoscale Distinctions Define Glutamatergic Synapses in Tonic vs Phasic Neurons. The Journal of neuroscience : the official journal of the Society for Neuroscience. 43:4598-4611.

Holderith, N., A. Lorincz, G. Katona, B. Rózsa, A. Kulik, M. Watanabe, and Z. Nusser. 2012. Release probability of hippocampal glutamatergic terminals scales with the size of the active zone. Nature neuroscience. 15:988-997.

Jetti, S.K., A.B. Crane, Y. Akbergenova, N.A. Aponte-Santiago, K.L. Cunningham, C.A. Whittaker, and J.T. Littleton. 2023. Molecular Logic of Synaptic Diversity Between *Drosophila* Tonic and Phasic Motoneurons. bioRxiv:2023.2001.2017.524447.

Kurshan, P.T., A. Oztan, and T.L. Schwarz. 2009. Presynaptic alpha2delta-3 is required for synaptic morphogenesis independent of its Ca2+-channel functions. Nature neuroscience. 12:1415-1423.

Lu, Z., A.K. Chouhan, J.A. Borycz, Z. Lu, A.J. Rossano, K.L. Brain, Y. Zhou, I.A. Meinertzhagen, and G.T. Macleod. 2016. High-Probability Neurotransmitter Release Sites Represent an Energy-Efficient Design. Current biology : CB. 26:2562-2571.

Ly , C.V., C.-K. Yao , P. Verstreken , T. Ohyama , and H.J. Bellen 2008. straightjacket is required for the synaptic stabilization of cacophony, a voltage-gated calcium channel α1 subunit. Journal of Cell Biology. 181:157-170.

Mrestani, A., M. Pauli, P. Kollmannsberger, F. Repp, R.J. Kittel, J. Eilers, S. Doose, M. Sauer, A.-L. Sirén, M. Heckmann, and M.M. Paul. 2021. Active zone compaction correlates with presynaptic homeostatic potentiation. Cell Reports. 37:109770.

Nakamura, Y., H. Harada, N. Kamasawa, K. Matsui, Jason S. Rothman, R. Shigemoto, R.A. Silver, David A. DiGregorio, and T. Takahashi. 2015. Nanoscale Distribution of Presynaptic Ca2+ Channels and Its Impact on Vesicular Release during Development. Neuron. 85:145-158.

Newman, Z.L., D. Bakshinskaya, R. Schultz, S.J. Kenny, S. Moon, K. Aghi, C. Stanley, N. Marnani, R. Li, J. Bleier, K. Xu, and E.Y. Isacoff. 2022. Determinants of synapse diversity revealed by superresolution quantal transmission and active zone imaging. Nature Communications. 13:229.

Rebola, N., M. Reva, T. Kirizs, M. Szoboszlay, A. Lőrincz, G. Moneron, Z. Nusser, and D.A. Digregorio. 2019. Distinct Nanoscale Calcium Channel and Synaptic Vesicle Topographies Contribute to the Diversity of Synaptic Function. Neuron. 104:693-710.e699.

Sheng, J., L. He, H. Zheng, L. Xue, F. Luo, W. Shin, T. Sun, T. Kuner, D.T. Yue, and L.-G. Wu. 2012. Calcium-channel number critically influences synaptic strength and plasticity at the active zone. Nature neuroscience. 15:998-1006.

Wang, T., R.T. Jones, J.M. Whippen, and G.W. Davis. 2016. alpha2delta-3 Is Required for Rapid Transsynaptic Homeostatic Signaling. Cell Rep. 16:2875-2888.

**Reviewer #1 (Recommendations For The Authors):**
Major points:(1) A central question regarding VGCC differences at Is vs Ib active zones is why is calcium influx higher at Is active zones compared to Ib. Ideally, the authors would have started this study by showing correlations between Cac abundance, presynaptic calcium influx, and Pr at Is vs Ib active zones. If they had, they would likely find that Cac abundance scales with calcium influx and Pr within Is vs Ib, but that calcium influx is over two-fold enhanced at Is over Ib when normalized to the same Cac abundance. This is more than sufficient to explain the Pr differences, so the rest of the study should have focused on revealing why influx is different at Is over Ib despite an apparently similar level of Cac abundance. Then the examination of CaBeta, Stj, etc could have been used to help explain this conundrumA lesson might be gleaned in how to structure this narrative from the Rebola 2019 study, which the authors cite and discuss at length. Similar to the current study, that paper started with two synapses ("strong" vs "weak") and sought to explain why they were so different in synaptic strength. First, they examined presynaptic calcium influx, and surprisingly found that the strong synapse had reduced calcium influx compared to the weak. Then the rest of the paper sought to explain why synaptic strength (Pr) was higher at the strong synapse despite reduced calcium influx. The authors do not use this logical flow and narrative in the present study, despite the focus being on how Cav2 channels contribute to strong vs weak synapses - and the primary function of Cav2 channels is to pass calcium at active zones to drive vesicle fusion.Although the authors did not show that presynaptic calcium influx is higher at Is vs Ib active zones in the current manuscript, other studies have previously established that calcium influx is two-fold higher at Is active zones vs Ib (as the authors cite). Rather than focusing so much on Pr at Is vs Ib active zones, which as the authors know can be influenced by myriad differences, it seems the more relevant parameter to study is simply to address presynaptic calcium influx at Is vs Ib, which is the primary function of Cac. Put more simply, if Cac levels are the same at Is vs Ib active zones, why is calcium influx at least two-fold higher at Is?It would therefore seem crucial for the authors to determine presynaptic calcium influx levels (ideally at individual AZs) to really understand how Cac intensity levels correlate with calcium influx. The authors instead map Pr at individual AZs, but as the authors know there are many variables that influence whether a SV releases in addition to calcium influx. There are a number of options for this kind of imaging in *Drosophila*, including genetically encoded calcium indicators targeted to active zones. But since several studies have previously established that influx is higher at Is active zones over Ib, this may not be necessary. That being said, there is a lot of value in quantitatively analyzing Cac/Stj/CaBeta abundance, calcium influx, and Pr together at individual active zones.

We appreciate the perspective that we could have focused on why Ca2+ influx is 2x greater at type Is active zones, which we agree is an important and interesting question. However, growing evidence indicates that Ca2+ influx alone, like Ca2+ channel abundance, does not reliably predict synaptic strength between inputs. So, here we focused instead on how other differences between synapses influence Pr and contribute to synaptic heterogeneity between and/or among synapses formed by strong and weak inputs. We have changed our title and framing to better reflect this focus.

As Reviewer 1 notes, Rebola et al. (2019) found that lower Pr granule synapses exhibit higher Ca2+ influx (and Ca2+ channel abundance). In another example, Aldahabi et al. (2022) demonstrated that even when Ca2+ influx is greater at high-Pr synapses, it does not necessarily explain differences in synaptic strength as raising Ca2+ entry at low-Pr synapses to high-Pr synapse levels was not sufficient to increase synaptic strength to high-Pr input levels. Similar findings have been reported at tonic and phasic synapses of the Crayfish NMJ (Msghina, 1999).

Several lines of evidence argue that factors beyond Ca2+ influx also play important roles in establishing distinct release properties at the *Drosophila* NMJ. A recent study using using a botulinum transgene to isolate type Ib and Is synapses for electrophysiological analysis found that increasing external [Ca^2+^] from physiological levels (1.8 mM) to 3 mM or even 6 mM does not result in a 3-fold increase in EPSCs or quantal content at type Ib synapses despite the prediction that the increase would be even greater given the power dependence of release on between Ca2+ concentration (He et al., 2023). The authors further found that type Ib synapses are more sensitive than type Is synapses to the slow Ca^2+^ chelator EGTA, indicating looser Ca2+ channel-SV coupling.

Consistently, we find that although VGCC levels are similar at the two inputs, their density is greater at type Is active zones (Figs. 1 and 2). Our findings also reveal additional molecular differences that may contribute to the observed differences in neurotransmitter release properties between the two inputs, including lower levels of the active zone protein Brp (Fig 3) and the auxiliary subunit α2δ-3/Stj (Fig. 6) at high Pr type Is inputs. In contrast, levels of each of these proteins positively correlate with synaptic strength among active zones of a single input, whether low- or high-Pr (Figs. 1, 3, 6). Similarly, levels of each of these proteins increase during homeostatic potentiation of neurotransmitter release (Figs. 4 and 7). Thus, we propose that two broad mechanisms contribute to synaptic diversity in the nervous system: (1) spatial organization and relative molecular content establish distinct average basal release probabilities that differ between inputs and (2) among individual synapses of distinct inputs, coordinated modulation of Ca^2+^ channel and active zone protein abundance independently tunes Pr. These intersecting mechanisms provide a framework for understanding the extensive and dynamic synaptic diversity observed across nervous systems.

(2) In addition to key points made above, it seems the authors should at least consider (if not experimentally test) what other differences might contribute to the higher calcium influx at Is over Ib:- Distinct splice isoforms of Cac (and/or Stj/Cabeta): The recent RNAseq analysis of gene expression at Is vs Ib motor neurons from Troy Littleton's group may inform this consideration?- Stj reduction at Is: Do channel studies in heterologous systems give any insight into VGCC channel function with and without a2d-3? Do Cav2 channels without a2d pass more calcium? This would then offer an obvious solution to the key conundrum underlying this study.

These are excellent questions that we are actively pursuing. While there is no evidence of differentially expressed splice isoforms of Stj or *Ca-β* in the recent RNA-seq data from Jetti et al., 2023, subtle changes in Cac isoform usage were observed that may contribute to differences in Ca2+ influx. In heterologous systems, α2δ expression generally increases Ca2+ channel membrane insertion and Ca2+ currents. However, in vivo α2δ’s can also mediate extracellular interactions that may modulate channel function. We address these points in greater detail in the revised discussion.

(3) Assess Stj and CaBeta levels at AZs after PhTx: The successful generation of endogenously tagged Stj and CaBeta enables some relatively easy experiments that would be of interest, similar to what the authors present for Cac. Does Brp similarly control Stj and CaBeta at Is vs Ib compared to what they show for Cac? In addition, does homeostatic plasticity similarly change Stj and CaBeta at Is vs Ib compared to what the authors have shown for Cac? i.e., do they both similarly increase in intensity, by the same amount, as Cac?

We agree and have included an analysis of α2δ-3/Stj levels following PhTx exposure (Fig. 7A-C). We have also investigated the regulation of Stj during chronic presynaptic homeostatic potentiation (Fig. 7D-F). In both cases, StjV5-N levels significantly increase at type Ib and Is active zones, consistent with our finding that among AZs of either type Ib or Is inputs, Stj levels correlate with Cac abundance and, thus, Pr. Together with our and others’ findings, this suggests that coordinated increases Ca2+ channel, auxiliary subunit, and active zone protein abundance positively tunes synaptic strength at diverse synaptic subtypes.

Minor points:(1) Including line numbers would make reviewing/commenting easier.

We apologize for this oversight and have added line numbers to the revised manuscript.

(2) Fig. 2I: It is not apparent what the mean cluster density is between Ib vs Is (as it is in Fig. 2F-H graphs). The mean and error bars should be included in 2I as it is in 2G. Same with Fig. 3C.

Thank you for pointing this out. We have added error bars to the paired analysis in 2I as well as in 3C and 1C.

(3) Fig. 4 - it might make more sense to normalize Brp and Cac intensity as a percentage of baseline (PhTx at Is or Ib) rather than normalizing everything to control Ib.

We have revised the graphs as suggested in Figure 4 and throughout.

(4) Page 5 bottom - REFS missing after Fig. 1E.

Thank you for catching this. We have fixed it.

**Reviewer #2 (Recommendations For The Authors):**
This reader found differentiating between low Pr sites (deep purple) and cac measurements (black) difficult in Fig 1B. You may consider depicting this differently.

Thank you for this feedback. We have changed the color scheme to improve readability.

I found it difficult to discern the difference between experiments Fig 1E and Fig 1J. Why are individual dots distributed differently?

The individual data points are the same as in 1E and 1F, but we have removed the individual NMJ dimensionality to combine all Is and Ib data points together along with best fit lines for comparison of their slopes. We have added text to the revised manuscript to clarify this.

Results section, second paragraph, add references, remove 'REF': We next investigated the correlation between Pr and VGCC levels and found that at type Is inputs, single-AZ Cac intensity positively correlates with Pr (Fig. 1E; REFS).

Thank you. We have corrected this error.